# Associations between preoperative Oxford hip and knee scores and costs and quality of life of patients undergoing primary total joint replacement in the NHS England: an observational study

Peter Eibich,[1,2] Helen A Dakin,[1] Andrew James Price,[3] David Beard,[3] Nigel K Arden,[3,4] Alastair M Gray[1]

[1]Health Economics Research Centre, Nuffield Department of Population Health, University of Oxford, Oxford, UK
[2]Max Planck Institute for Demographic Research, Rostock, Germany
[3]Nuffield Department of Orthopaedics, Rheumatology and Musculoskeletal Sciences, University of Oxford, Oxford, UK
[4]Faculty of Medicine, University of Southampton, Southampton, UK

**Correspondence to**
Dr Peter Eibich;
eibich@demogr.mpg.de

## ABSTRACT

**Objectives** To assess how costs and quality of life (measured by EuroQoL-5 Dimensions (EQ-5D)) before and after total hip replacement (THR) and total knee replacement (TKR) vary with age, gender and preoperative Oxford hip score (OHS) and Oxford knee score (OKS).

**Design** Regression analyses using prospectively collected data from clinical trials, cohort studies and administrative data bases.

**Setting** UK secondary care.

**Participants** Men and women undergoing primary THR or TKR. The Hospital Episode Statistics data linked to patient-reported outcome measures included 602 176 patients undergoing hip or knee replacement who were followed up for up to 6 years. The Knee Arthroplasty Trial included 2217 patients undergoing TKR who were followed up for 12 years. The Clinical Outcomes in Arthroplasty Study cohort included 806 patients undergoing THR and 484 patients undergoing TKR who were observed for 1 year.

**Outcome measures** EQ-5D-3L quality of life before and after surgery, costs of primary arthroplasty, costs of revision arthroplasty and the costs of hospital readmissions and ambulatory costs in the year before and up to 12 years after joint replacement.

**Results** Average postoperative utility for patients at the 5th percentile of the OHS/OKS distribution was 0.61/0.5 for THR/TKR and 0.89/0.85 for patients at the 95th percentile. The difference between postoperative and preoperative EQ-5D utility was highest for patients with preoperative OHS/OKS lower than 10. However, postoperative EQ-5D utility was higher than preoperative utility for all patients with OHS≤46 and those with OKS≤44. In contrast, costs were generally higher for patients with low preoperative OHS/OKS than those with high OHS/OKS. For example, costs of hospital readmissions within 12 months after primary THR/TKR were £740/£888 for patients at the 5th percentile compared with £314/£404 at the 95th percentile of the OHS/OKS distribution.

**Conclusions** Our findings suggest that costs and quality of life associated with total joint replacement vary

### Strengths and limitations of this study

► This study used data from different sources, including clinical trials, cohort studies and administrative databases.
► The sample sizes are considerably larger than those used in previous studies.
► The optimal regression models were identified in a systematic model selection process designed to maximise predictive accuracy.
► Our largest dataset contained no information on potentially important variables such as body mass index.

systematically with preoperative symptoms measured by OHS/OKS.

## INTRODUCTION

Total joint replacement (TJR) has been shown to greatly improve the quality of life of patients with osteoarthritis.[1–5] Moreover, previous studies have demonstrated that both total knee replacement (TKR) and total hip replacement (THR) are highly cost-effective.[3 4] Nevertheless, in the UK access to TJR surgery is frequently rationed based on patient characteristics (such as age and functional status)[6] since it is a non-urgent procedure with a high impact on healthcare budgets. In particular, the Oxford hip score (OHS) and Oxford knee score (OKS) were used in certain regions to define threshold values (varying from 19 to 30) above which patients are not eligible for TJR.[7–10] This is based on the argument that patients with lower preoperative functional status benefit more from TJR, but the evidence to support

this is limited, and very few studies have examined how outcomes or costs of TJR differ by preoperative patient characteristics.[1–5 11–16]

Most of these studies focused on demographic characteristics such as age, with only five studies exploring the impact of preoperative function.[3 4 13 14 16] Cushnaghan et al[13] examined the associations between baseline physical functioning (measured by 36-Item Short Form Health Survey (SF-36)) and the long-term outcome of TKR (measured by the change in SF-36). Judge et al[16] investigated the associations between preoperative OHS and postoperative OHS and concluded that all patients benefited substantially from THR. Only Dakin et al[4] and Fordham et al[3] investigated the impact of OKS and OHS on costs and outcomes of TKR and TJR, respectively, and both studies concluded that while health benefits and costs do vary with preoperative OHS/OKS score, TJR improves quality of life for almost all patients.

One recent US study by Ferket et al[14] concluded that TKR results in only minimal health gains for patients with good physical function (measured by the SF-12 physical score), and that, on economic grounds, access to TKR should be restricted to patients with severe symptoms; however, this study was based on only 382 patients undergoing TKR. Other studies in this area have also relied on small datasets (eg, 2138 patients undergoing TKR[4] or 938 patients undergoing THR[3] followed for 5 years), which are likely to contain very few patients with low or high scores. The results from these studies are therefore subject to considerable uncertainty.

This study aimed to estimate the relationship between preoperative patient characteristics (age, sex and OHS/OKS) and the costs and quality of life of patients undergoing TJR in England. We used administrative data from the National Health Service (NHS) patient-reported outcome measures (PROMs) programme linked to Hospital Episode Statistics (HES) data, alongside smaller datasets from the Knee Arthroplasty Trial (KAT) and the Clinical Outcomes in Arthroplasty Study (COASt) cohort study to obtain reliable estimates on a wide range of outcomes and patients, including for patients with very high OHS/OKS.

## METHODS
### Datasets
The study focused on patients undergoing elective primary hip or knee arthroplasty from three data sources: the KAT trial,[17] the COASt cohort study[18] and data from the NHS PROMs programme linked to HES data.[19] These datasets were chosen from a wider range of available candidate datasets[3 20–22] based on the availability of relevant outcome and baseline data as well as sample size considerations.

The NHS PROMs programme routinely collects data on all patients undergoing hip and knee replacement on the NHS in England.[23] Data are collected via questionnaires at baseline (typically the day of admission for surgery) and

at minimum 6 months post surgery. The dataset used in this study covered the period from April 2009 to October 2015. We linked these data to Admitted Patient Care data from HES,[19] which contains routine data on all patients treated in hospitals in the NHS England, with a particular focus on diagnoses and procedures. This yielded a sample of up to 602 176 patients who were followed up for up to 6 years. The data include patients undergoing primary joint replacement surgery other than TJR (eg, hip resurfacing) since we were not able to reliably identify the type of operation from the available data.

KAT is a randomised controlled trial comparing different types of knee prosthesis.[17] We used data on EuroQoL-5 Dimensions (EQ-5D) utility and ambulatory costs following 2217 UK patients, who were followed up annually for up to 12 years after TKR.

COASt[18] is a prospective, dual-centre longitudinal cohort study recruiting patients across two hospitals: University Hospital Southampton NHS Foundation Trust and Nuffield Orthopaedic Centre (part of Oxford University Hospitals NHS Trust). COASt was established in 2010 and recruited patients placed on the waiting list for knee or hip replacement surgery. The datasets extracted for this study contained observations for 810 patients undergoing hip surgery and 858 patients undergoing knee surgery. After excluding patients who underwent procedures other than THR/TKR (eg, partial knee replacement or hip resurfacing), the datasets contained 806 patients undergoing THR and 484 patients undergoing TKR. Data are collected prior to surgery as well as at 6 weeks and then annually for 5 years thereafter, although only the first year of data were used in this study.

### Outcomes
#### Quality of life
Preoperative and postoperative quality of life was measured using EQ-5D-3L, valued using the UK time trade-off tariff.[24] EQ-5D-3L health utility takes values between −0.594 and 1, with higher values representing higher quality of life. We chose to focus on EQ-5D since it is the preferred measure of quality of life for economic evaluations submitted to the National Institute for Health and Care Excellence.

Linked NHS PROMs/HES data were used to estimate quality of life before surgery and 6 months after THR or TKR. Data from KAT were used to estimate annual changes in quality of life between 6 months and 12 years after TKR.

#### Costs
#### *Primary and revision arthroplasty*
We estimated the impact of OHS/OKS on the costs of primary and revision TJR using linked NHS PROMs/HES data. We identified HES observations related to primary or revision arthroplasty by linking NHS PROMs data to the HES database, and then excluded all observations marked as revisions in the corresponding NHS PROMs questionnaire. If a patient was observed to have the same

procedure (THR or TKR) carried out at the same joint (right-side or left-side) more than once, we assumed that the earliest observed procedure was a primary surgery episode, while all subsequent procedures were revision episodes. Similarly, we identified revisions as those episodes that were either marked as a revision in the NHS PROMs data or where we observed an earlier procedure on the same joint in the HES data. This implies that for patients undergoing primary joint replacement other than TJR first and TJR on the same joint later, the first procedure would be coded as primary surgery and the second procedure would be coded as a revision since we were not able to reliably identify the type of operation based on the available data.

### Readmissions after primary and revision arthroplasty

We derived the costs of hospital readmissions after primary arthroplasty using PROMs/HES data. Our HES data extract included all hospital admissions for patients contained in the NHS PROMs data (ie, who underwent primary or revision arthroplasty between April 2009 and October 2015). In a first step, we split the data into two datasets, one consisting of all 'index' (ie, primary or revision arthroplasty) episodes and one containing all other episodes. Then we matched the non-index admissions to the index episodes. If a hospital admission could be matched to more than one index episode, we matched the admission to the closest arthroplasty episode that occurred before the hospital admission. However, we excluded arthroplasty episodes from the match if the hospital admission referred to a different joint or side of the body than the arthroplasty episode.

To determine whether a matched hospital admission can be attributed to primary or revision arthroplasty, we applied the following set of criteria:

► The admission occurred within 30 days after primary arthroplasty.
► The admission had a primary diagnosis for hip or knee arthritis.
► The admission had a procedure code referring to the hip or knee joint.
► The admission had a diagnosis code associated with infections of the skin, the joint or the prosthesis.

These criteria were developed based on discussions with a clinical registrar and a clinical coding manager. The detailed diagnosis and procedure codes are available in the online supplementary file (section A). Any admission that met at least one of the criteria was retained as a relevant episode.

Finally, we aggregated all readmissions by year from primary arthroplasty to create an annual measure of readmissions for each observed primary arthroplasty procedure. We assumed that readmissions were zero for those patients known to be alive with no recorded admissions in any given year. The resulting dataset contains observations for all patients (including those without hospital readmissions) and all years for up to 6 years following primary arthroplasty. Patients were considered to be observed

either until the end of the study period (October 2015) or until their death if they died in hospital. Deaths outside of the hospital were not observed in the data. We analysed readmission costs in the first 12 months after primary arthroplasty separately from annual readmission costs between 1 and 6 years after surgery. We also conducted a separate analysis of readmission costs in the year of revision surgery and annual readmission costs after revision surgery.

### Ambulatory costs after primary arthroplasty and costs before primary arthroplasty

Ambulatory costs included outpatient and community costs related to primary arthroplasty, for example, general practitioner visits, nurse visits, physiotherapy, outpatient appointments at a hospital or visits to the 'Accidents & Emergency' department. We excluded costs such as medication, nursing home care, personal care and equipment or home modifications due to a lack of data.

For THR, data on ambulatory costs in the first 12 months after primary arthroplasty were obtained from the COASt study, but annual ambulatory costs >12 months after primary hip arthroplasty could not be analysed due to a lack of data. For TKR, we used data on annual ambulatory costs from the KAT trial for up to 12 years following primary knee arthroplasty and derived separate estimates for costs in the first 12 months after TKR and annual costs between 1 and 12 years after TKR. Data on nurse visits and visits to the 'Accidents & Emergency' department were not available from KAT, which means that ambulatory costs after TKR might be slightly underestimated. For TKR, we also conducted a separate analysis of ambulatory costs in the year of revision surgery and in subsequent years.

Finally, we analysed costs in the year before primary arthroplasty based on data from COASt. We included both ambulatory costs (as above) and hospital admissions.

### Costing

We chose 2014 as the reference year for costs since this is the most recent year which is completely observed in our data extract.

Hospital admissions in the HES data were valued using the NHS Local Payment Grouper 2014/2015[25] and prices from the 2014/2015 National Schedule.[26] For readmissions, we distinguished between elective and non-elective admissions and used the respective prices. For those hospital readmissions that could not be costed (eg, due to errors in the grouping process or missing data on length of stay), we imputed costs by using either the mean cost per bed-day for admissions in the same category (if data on length of stay were available), or (if data on length of stay were missing) by using the mean cost per admission that meet the same criteria as a relevant readmission episode. The imputed values are provided in online supplementary table B.1. We did not impute costs for

primary and revision arthroplasty episodes and omitted observations that could not be costed.

Costs for ambulatory visits and hospital admissions in KAT and COASt were derived by attaching the relevant unit costs for 2014 (see online supplementary table B.1) to the number of visits/appointments.

## Covariates

The OHS and OKS are PROMs, each consisting of 12 questions about pain and functional limitations specific to the joint. The total score varies between 0 and 48, with higher values indicating less pain and functional limitations.[27 28] We used OHS/OKS scores measured before primary arthroplasty in all models.

We also included age and sex in the regression models. When studying the impact of OHS/OKS on costs of primary arthroplasty, quality of life, costs before arthroplasty and costs in the first 12 months following arthroplasty, we included age at operation in the regression models. When studying the impact of OHS/OKS on costs >12 months after primary arthroplasty, we also considered including current age instead of age at operation as well as a time trend into the model. While the inclusion of further covariates (eg, body mass index or the American Society of Anesthesiologists physical status classification) would be desirable, this information was unfortunately not available in the NHS PROMs and HES datasets.

## Descriptive statistics

Table 1 provides an overview of all outcomes for which regression models were developed. Average postoperative EQ-5D utility was considerably higher than average preoperative utility. Interestingly, average EQ-5D utility before revision surgery was similar to or even lower than average preoperative utility. Average utility after revision was higher than average utility values before revision; however, it was lower than the average postoperative utility. On average, primary hip arthroplasty costed £5522 and primary knee arthroplasty costed £6053. Average cost in the 12 months before arthroplasty were £444 for THR and £836 for TKR. In the first 12 months after THR, average costs were £101 for ambulatory services and £455 for hospital readmissions. For TKR, average costs in the first 12 months after surgery were £361 for ambulatory services and £550 for hospital readmissions. Average preoperative OHS scores varied between 16.8 and 18.7 (due to the different samples), while average preoperative OKS scores varied between 16.9 and 19.4. For the sake of brevity, we discuss the modelling results for selected outcomes in this paper; the models for all other outcomes are provided in the online supplementary appendix.

## Statistical methods

The regression models presented in this study focus on predictive accuracy rather than causal inference. To this end, we developed a model selection procedure to systematically identify the best performing model from a range of candidate models. We began by conducting an exploratory data analysis for each outcome of interest to identify a range of candidate models. We visually inspected the distribution of each outcome as well as plots of the outcome against OHS/OKS and age. Based on these figures (available on request), we identified candidates for (i) the statistical model class (eg, ordinary least squares or Tobit regression), (ii) the functional form for OHS/OKS (eg, linear or quadratic trends), (iii) the functional form of the age trend and (iv) functional forms for the time trend (where applicable). In principle, any combination of these four components identifies a valid candidate model. However, in practice it was not feasible to test all possible candidate models against each other. Therefore, we selected our final model in four steps:

► In the first step, we selected the most appropriate model class from among the candidate models. All models included a linear trend for OHS/OKS, a linear trend for age and an indicator for sex of the patient and models on longitudinal data also included a linear trend for time since arthroplasty. Depending on the exploratory data analysis, we considered linear regression models, generalised linear models, two-part and Tobit regression models for censored data.

► Where applicable, we then chose the functional form for the time trend using the model class identified in the first step.

► In the second step, we chose the functional form for OHS/OKS. We used the model class identified in the first step and included a linear trend for age as well as an indicator for sex of the patient. We considered polynomials (eg, linear, quadratic or cubic polynomials) as well as linear splines, where the spline points were identified in the exploratory data analysis.

► The third step consisted of choosing the functional form for age. We used the model class identified in the first step and the functional form for OHS/OKS from the second step. We considered polynomial trends, linear splines as well as age categories, depending on the exploratory data analysis. We also considered whether excluding age would improve prediction accuracy. When modelling costs >12 months after primary arthroplasty, we first examined whether age at operation or current age resulted in higher predictive accuracy using linear trends before choosing the functional form for the chosen age variable.

► In the fourth and last step, we evaluated an alternative model selected in the third step that omitted the indicator for sex of the patient, to assess whether this improved prediction accuracy.

We used mean squared error (MSE) as our criterion for model selection and used a 10-fold cross-validation process to avoid overfitting. We randomly divided our sample into 10 slices. For each of the steps outlined above, we estimated the candidate models using nine of these data slices, then predicted the outcomes for the slice left out of the estimation, and finally calculated the MSE of the prediction. This was repeated 10 times, so that each data slice was used once for prediction (ie, internal validation) and the squared

**Table 1**  Data sources and sample sizes

| Variable of interest | Dataset | Mean±SD (dependent variable) | Mean OHS | Final sample size | Missing covariates (%) | Table | Figure |
|---|---|---|---|---|---|---|---|
| **A. Quality of life for THR** | | | | | | | |
| Postoperative EQ-5D utility | PROMs-HES | 0.784±0.247 | 18.1 | 208 345 | 0.96 | C.1 | 1A |
| Preoperative EQ-5D utility | PROMs-HES | 0.334±0.324 | 17.6 | 271 045 | 0.83 | C.2 | 1A |
| EQ-5D utility before revision surgery | PROMs-HES | 0.348±0.341 | 17.2 | 1391 | 1.21 | C.4 | – |
| EQ-5D utility after revision surgery | PROMs-HES | 0.648±0.316 | 17.7 | 884 | 0.90 | C.5 | – |
| **B. Costs (in £) for THR** | | | | | | | |
| Costs of primary arthroplasty | PROMs-HES | 5,522±744 | 17.5 | 286 507 | 1.12 | C.6 | 1C |
| Costs of revision arthroplasty | PROMs-HES | 7,994±2362 | 16.9 | 2359 | 1.17 | C.7 | 1E |
| Costs in the 12 months before arthroplasty | COASt | 444±679 | 18.7 | 441 | 16.00 | C.8 | 2A |
| Ambulatory costs<12 months after primary arthroplasty | COASt | 101±244 | 18.5 | 548 | 16.72 | C.9 | 2E |
| Readmissions<12 months after primary arthroplasty | PROMs-HES | 455±2150 | 17.5 | 236 514 | 1.48 | C.10 | 2C |
| Annual cost of readmissions between 1 and 6 years after primary arthrolasty | PROMs-HES | 118±1137 | 17.6 | 476 514 | 0.98 | C.11 | 3A |
| Readmissions in the year of revision surgery | PROMs-HES | 780±3507 | 16.8 | 1669 | 1.07 | C.12 | 3C |
| Annual cost of readmissions after revision surgery | PROMs-HES | 148±1309 | 17.0 | 2406 | 0.87 | C.13 | 3E |
| **C. Quality of life for TKR** | | | | | | | |
| Postoperative EQ-5D utility | PROMs-HES | 0.723±0.259 | 19.0 | 223 836 | 0.96 | C.1 | 1B |
| Preoperative EQ-5D utility | PROMs-HES | 0.391±0.317 | 18.4 | 290 983 | 0.98 | C.2 | 1B |
| EQ-5D utility between 1 and 12 years after primary arthroplasty | KAT | 0.693±0.276 | 18.4 | 15 414 | 0.00 | C.3 | – |
| EQ-5D utility before revision surgery | PROMs-HES | 0.291±0.331 | 17.0 | 2227 | 1.07 | C.4 | – |
| EQ-5D utility after revision surgery | PROMs-HES | 0.544±0.340 | 17.7 | 1398 | 0.99 | C.5 | – |
| **D. Costs (in £) for TKR** | | | | | | | |
| Costs of primary arthroplasty | PROMs-HES | 6,053±1097 | 18.4 | 308 638 | 1.21 | C.6 | 1D |
| Costs of revision arthroplasty | PROMs-HES | 7,759±2777 | 16.9 | 3416 | 1.13 | C.7 | 1F |
| Costs in the 12 months before arthroplasty | COASt | 836±2131 | 19.4 | 278 | 10.03 | C.8 | 2B |
| Ambulatory costs<12 months after primary arthroplasty | KAT | 361±400 | 18.3 | 1841 | 3.21 | C.9 | 2F |
| Readmissions<12 months after primary arthroplasty | PROMs-HES | 550±2472 | 18.4 | 255 194 | 1.22 | C.10 | 2C |
| Annual cost of readmissions between 1 and 6 years after primary arthrolasty | PROMs-HES | 176±1311 | 18.4 | 514 047 | 1.18 | C.11 | 3B |
| Readmissions in the year of revision surgery | PROMs-HES | 844±3361 | 16.9 | 2258 | 1.05 | C.12 | 3D |
| Annual cost of readmissions after revision surgery | PROMs-HES | 246±1877 | 16.9 | 3153 | 1.10 | C.13 | 3F |
| Annual ambulatory costs between 1 and 12 years after primary arthroplasty | KAT | 29±104 | 18.4 | 13 271 | 4.13 | C.14 | – |
| Ambulatory costs in the year of revision surgery | KAT | 578±718 | 17.8 | 88 | 4.35 | C.15 | – |
| Annual ambulatory costs after revision surgery | KAT | 129±235 | 18.1 | 329 | 3.24 | C.16 | – |

Tables C.1–C.16 are shown in the online supplementary file. 'Missing covariates' shows the fraction of observations that were excluded from the final sample due to missing data on one of the covariates (including OHS/OKS) included into the model.
COASt, Clinical Outcomes in Arthroplasty Study; EQ-5D, EuroQoL-5 Dimensions; HES, Hospital Episode Statistics; KAT, Knee Arthroplasty Trial; OHS, Oxford hip score; OKS, Oxford knee score; PROM, patient-reported outcome measure; THR, total hip replacement.

error was calculated for each candidate model and for every observation in the sample. Then, we averaged across all patients (in all 10 slices) to estimate the MSE for each candidate model and chose the model with the lowest MSE. Finally, we re-estimated the model using the whole dataset to obtain our final estimates.

All regression models were estimated using Stata V.14 (StataCorp). We conducted a complete-case analysis, omitting observations with missing data on any of the variables of interest. For the model selection procedure, we omitted all observations with missing data on outcome, OHS or OKS, age as well as sex to ensure that all candidate models were evaluated based on the same dataset. For the final estimation, we only omitted observations with missing data on any of the included variables. With the exception of the models estimated on COASt data, very few observations were excluded due to missing information on covariates (table 1). We visualise the estimated associations between OHS/OKS and quality of life as well as costs by plotting predicted costs and quality of life against OHS/OKS scores. We generated predictions using the estimation sample, that is, for each OHS/OKS score we averaged predictions over the sample distributions of the other covariates. The full estimation results as well as the candidate models considered for each outcome are provided in online supplementary file (section C). Online supplementary file (section D) provides details and examples for calculating predictions from the different models used in this study.

## RESULTS
### Quality of life

Figure 1A,B shows the relationship between OHS/OKS and preoperative as well as postoperative EQ-5D utility. The lines show the fitted values, while the grey areas represent 95% CIs. Both preoperative and postoperative quality of life were higher for patients with high OHS/OKS scores (figure 1A, B). However, the relationship is clearly non-linear. Preoperative quality of life increased considerably from −0.17/−0.16 for OHS/OKS of 0 to 0.59/0.61 for OHS/OKS scores of 25, beyond which the

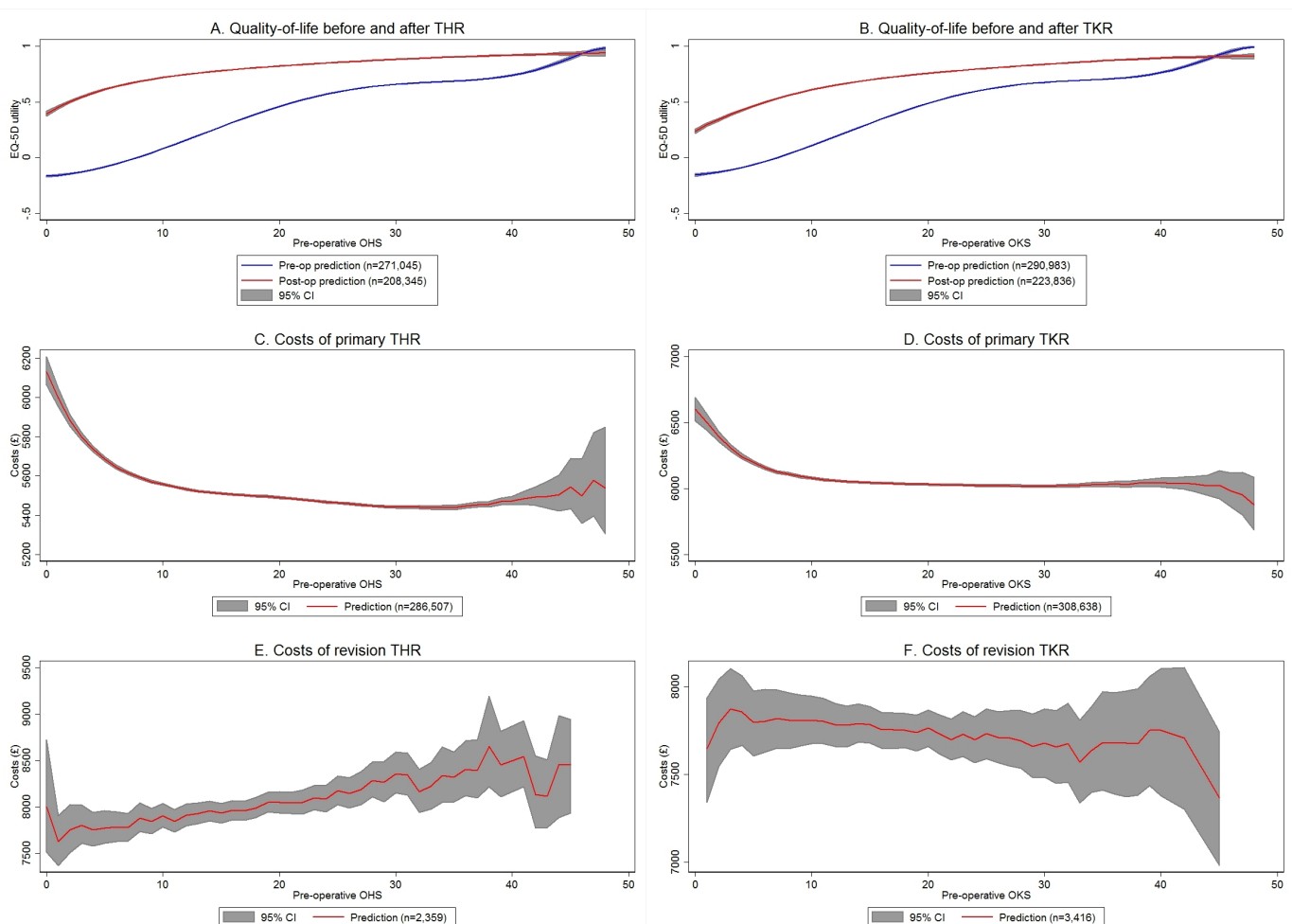

**Figure 1** Associations between OHS/OKS and quality-of-life, costs of primary and revision TJR. Costs are measured in 2014 pound sterling (£). Lines show predicted values averaged over all observations in the sample with a given Oxford hip score (OHS)/Oxford knee score (OKS). All models included preoperative OHS or OKS score, age at operation and sex of the patient as independent variables, with the exception of the model for preoperative EQ-5D of total hip replacement (THR) patients, which did not include an indicator for female patients. Full regression results are shown in online supplementary appendix C. The grey areas show 95% CIs obtained from 1000 bootstrap replications. National Health Service patient-reported outcome measures and Hospital Episode Statistics data, 2009– 2015; own calculations. EQ-5D, EuroQoL-5 Dimensions; TJR, total joint replacement; TKR, total knee replacement.

increase in quality of life by OHS/OKS was much smaller. In contrast, postoperative quality of life increased strongly from 0.39/0.24 for OHS/OKS of 0 to 0.72/0.61 for scores of 10, after which there were only small differences in quality of life by OHS/OKS. As a consequence, the health gain associated with TJR (ie, the difference between the two curves) was largest for patients with low scores (0.69 for OHS of 5 and 0.53 for OKS of 6). While the models predicted that postoperative EQ-5D utility was higher than preoperative utility for all patients with OHS≤46 and those with OKS≤44, the differences between postoperative and preoperative utility were marginal (0.005 for OHS of 46 and 0.02 for OKS of 44).

There was some variation in postoperative EQ-5D by age and sex: postoperative utility was highest for patients undergoing THR aged 60–69, and increased with age for patients aged 30–60 and then decreased with age for patients aged >70. Postoperative utility for patients undergoing TKR increased with age for all patients. Compared with men, women had higher preoperative utility before TKR (P<0.001), but lower postoperative utility after THR (P<0.001). Sex was excluded in the models for utility before THR and utility after TKR.

## Costs of primary and revision arthroplasty
The cost of primary arthroplasty decreased with OHS/OKS for scores between 0 (£6132 for THR and £6603 for TKR on average) and 10 (£5559 for THR and £6080 for TKR on average) (figure 1C, D). For scores between 10 and 40, the costs remained approximately constant. For scores >40, the predicted costs changed only slightly, while the uncertainty around these estimates increased considerably since <0.4% of the patients in the sample had a score >40. Costs for primary THR were lower (£22 on average) for women than for men (P<0.001), while the costs of TKR were higher (£9 on average) for women (P=0.026).

Figure 1E, F shows the relationship between the cost of revision arthroplasty and preoperative OHS/OKS score. Due to the smaller number of revisions, the uncertainty around these estimates is larger than for the costs of primary surgery, but the estimates suggest that the costs of THR revision surgery increased on average by £15 for a one-point increase in OHS (P=0.016), while the costs of TKR revision surgery remained constant (P=0.837 on the linear trend). The cost of revision arthroplasty was lower for older patients. Costs for THR revision surgery were on average £429 lower for women than for men (P<0.001).

## Costs in the year before TJR and in the first 12 months after surgery
Costs in the year before THR decreased significantly with increasing OHS (eg, £579 at OHS of 5 and £320 at OHS of 32; P=0.003), while costs before TKR varied little between OKS values of 0 and 20 (£897 at OKS of 10, £886 at OKS of 20), before decreasing thereafter (£513 at OKS of 32) (figure 2A, B) However, for both THR and TKR the curves are relatively flat. Costs decreased (although

insignificantly) with age for both THR and TKR. Since there was only one patient with an OHS of 33, who was relatively young, the age effect produced an apparent spike in costs for an OHS score of 33.

Figure 2C, D shows the estimated association between OHS/OKS and ambulatory costs in the first 12 months after TJR. Overall, ambulatory costs after THR were considerably lower than costs after TKR. A sensitivity analysis estimating ambulatory costs for patients undergoing TKR in COASt confirmed this finding, suggesting that this trend reflects the differences between joints rather than between datasets. Ambulatory costs decreased with increasing OHS/OKS scores from £51/£446 at OHS/OKS scores of 5 and 6, respectively, to £46/£286 at OHS/OKS scores of 32. The negative predicted costs in figure 2C are an artefact of the simple linear regression model, which does not restrict the outcome to be non-negative. It should be noted that the estimates for figure 2A–D are based on substantially smaller numbers than previous figures and are subject to considerable uncertainty.

## Costs of hospital readmissions
In line with the findings for the costs of primary surgery, the cost of hospital readmissions in the first 12 months after TJR decreased steeply with OHS/OKS for scores between 0 (£1543 for THR, £1700 for TKR) and 10 (£545 for THR, £684 for TKR), while for scores >10 the curve was almost flat (figure 2E, F). Both for THR and TKR, the costs were lower for women than for men (on average £78 for THR and £133 for TKR; P<0.001 for both models).

Figure 3 shows the relationship between OHS/OKS and the annual costs of hospital readmissions between 1 and 6 years after primary arthroplasty. The costs of hospital readmissions per year for patients without revision surgery are shown in figure 3A, B. For both THR and TKR, the costs decreased with OHS/OKS (£161/£266 for OHS/OKS scores of 5 and 6, respectively, £85/£120 for OHS/OKS scores of 32; P<0.001). For THR, the costs in year 2 (ie, between 12 and 24 months after THR) were significantly (on average £32; P<0.001) higher than in subsequent years. For TKR, the time trend indicates that readmission costs decreased over time (on average £31 per year; P<0.001).

Hospital readmission costs in the year of revision surgery did not have a statistically significant association with OHS/OKS scores measured before primary arthroplasty (figure 3C, D).

Finally, the costs of hospital readmissions per year for patients who had undergone revision surgery>12 months previously (figure 3E, F) were higher for patients who had low OHS/OKS scores before primary arthroplasty (P<0.05). However, the difference between patients with very low and very high scores was relatively small.

Table 2 summarises the predicted quality of life and costs for patients with selected OHS/OKS scores averaged over the sample distribution of age and sex. Differences in quality of life between patients with very high scores (95th percentile) and very low scores (5th percentile) were very

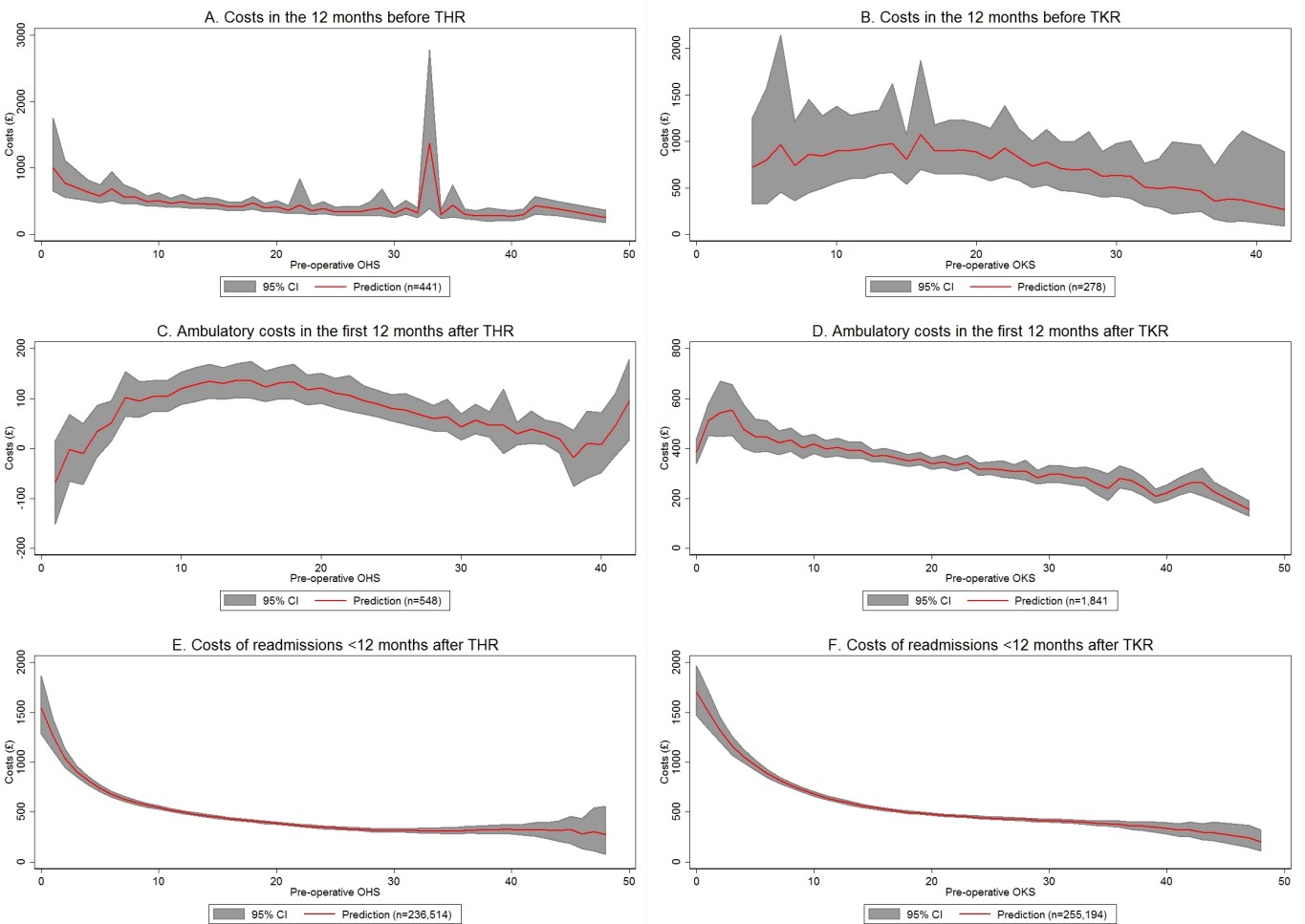

**Figure 2** Associations between OHS/OKS and costs before TJR, ambulatory and hospital readmission costs <12 months after TJR. Costs are measured in 2014 pound sterling (£). Lines show predicted values averaged over all observations in the sample with a given Oxford hip score (OHS)/Oxford knee score (OKS). All models included preoperative OHS or OKS score and age at operation as independent variables. The models for readmission costs also included a binary indicator for female patients. Full regression results are shown in online supplementary appendix C. The grey areas show 95% CIs obtained from 1000 bootstrap replications. National Health Service patient-reported outcome measures, Hospital Episode Statistics, Knee Arthroplasty Trial and Clinical Outcomes in Arthroplasty Study data, 2009–2015; own calculations. THR, total hip replacement; TJR, total joint replacement; TKR, total knee replacement.

large (0.75/0.72 point difference in preoperative EQ-5D utility and 0.28/0.35 point difference in postoperative utility for THR/TKR). Differences in costs were moderate in comparison. The costs of primary arthroplasty differed by £245 and £136 for THR and TKR, respectively. Costs in the year before surgery differed by £259 and £450, respectively. The cost of hospital readmissions in the first year differed by £427 and £484 for THR and TKR, while annual costs for hospital readmissions in years 1–6 after primary arthroplasty differed by £76 and £146 for patients undergoing THR and TKR.

Table 3 shows predicted average quality of life and costs for patients in different age groups. Associations between quality of life and age were weak overall, the largest difference between patients aged ≥80 and patients aged <60 was predicted for utility after revision TKR revision surgery (0.1 point difference). Differences in costs were more pronounced, with differences

in primary arthroplasty costs of £343 and £316 for THR and TKR. However, in contrast to table 2 the differences were not systematic. For example, costs of primary arthroplasty, readmission costs in the first 12 months and readmission costs in the year of revision surgery were higher for older patients, while other costs (eg, ambulatory costs in the first 12 months) were lower for older patients.

## DISCUSSION

OHS and OKS were developed as outcome measures reported by patients undergoing THR and TKR. Previous studies found that OHS and OKS are correlated with both clinical scores (such as the Charnley score or the American Knee Society score) and patient-reported general health measures (such as SF-36).[27 28] Thus, it is not surprising that we found that higher scores were systematically

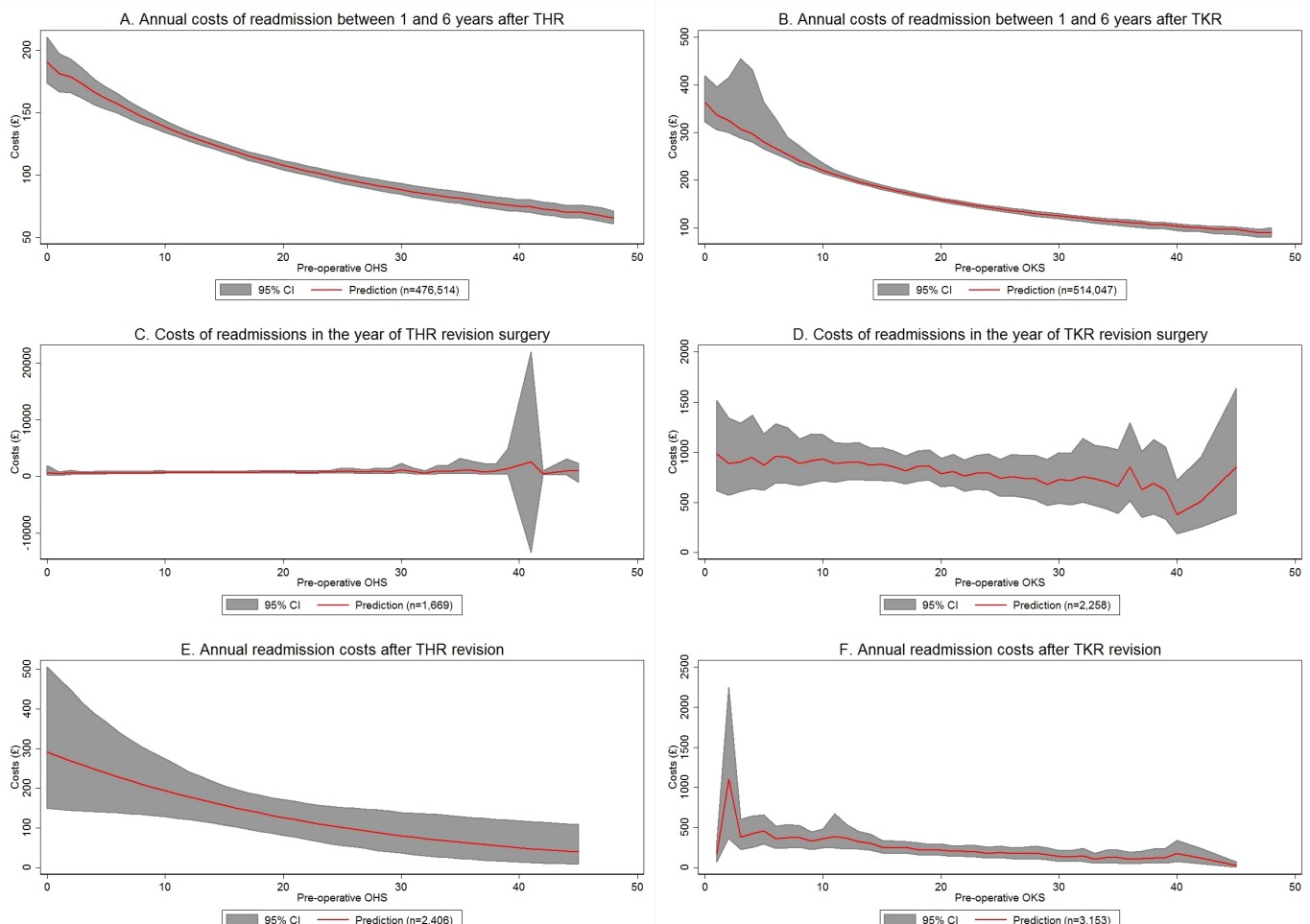

**Figure 3** Associations between OHS/OKS and hospital readmission costs >12 months after THR. Costs are measured in 2014 pound sterling (£). Lines show predicted values averaged over all observations in the sample with a given Oxford hip score (OHS)/Oxford knee score (OKS). All models included preoperative OHS or OKS score as independent variables. The model for annual readmission costs between 1 and 6 years after total hip replacement (THR) additionally included current age and an indicator for time since primary surgery. The corresponding model for total knee replacement (TKR) included age at operation, time since primary surgery and an indicator for sex of the patient as independent variables. The models in (B) additionally included age at operation, time since primary surgery and sex of the patient as covariates. The model for annual readmission costs after revision TKR surgery included current age and time since primary surgery as additional covariates. Full regression results are shown in online supplementary appendix C. The grey areas show 95% CIs obtained from 1000 bootstrap replications. National Health Service patient-reported outcome measures and Hospital Episode Statistics data, 2009–2015; own calculations.

associated with higher quality of life. However, the difference between postoperative and preoperative EQ-5D utility decreased with OHS/OKS. Patients with low scores have worse symptoms and more functional limitations due to osteoarthritis. Therefore, they have more potential to benefit from primary arthroplasty.

We also found that costs are moderately higher for patients with low preoperative scores. Patients with worse symptoms due to osteoarthritis might require longer recovery times. In addition, they are likely to have comorbidities or complications, which also increase the costs of primary arthroplasty. While the differences in costs are moderate compared with differences in quality of life (between £100 and £500 for most outcomes), it is worth noting that some of these costs are incurred annually, and

therefore even moderate differences can be economically important in the long run.

Our findings do not allow us to draw conclusions about the cost-effectiveness of TJR for patients with different preoperative scores without further assumptions about the costs and quality of life for patients without TJR. Nevertheless, based on these findings it appears promising to investigate the differences in cost-effectiveness as well as the impact of rationing policies. In particular, the finding that the associations between preoperative score and costs as well as quality of life are non-linear suggests that there could be a threshold score beyond which TJR might not be considered beneficial or cost-effective. However, identifying this threshold is beyond the scope of this paper and left to future research.

**Table 2** Predicted quality of life and costs by preoperative Oxford hip score (OHS)/Oxford knee score (OKS) at primary operation

| Outcome | THR | | | TKR | | |
|---|---|---|---|---|---|---|
| | OHS 5 (5th percentile) | OHS 17 (50th percentile) | OHS 32 (95th percentile) | OKS 6 (5th percentile) | OKS 18 (50th percentile) | OKS 32 (95th percentile) |
| A. Quality of life | | | | | | |
| Postoperative EQ-5D utility | 0.612 | 0.798 | 0.890 | 0.498 | 0.735 | 0.851 |
| Preoperative EQ-5D utility | −0.082 | 0.354 | 0.671 | −0.033 | 0.421 | 0.687 |
| EQ-5D utility>12 months after primary arthroplasty | – | – | – | 0.501 | 0.700 | 0.813 |
| EQ-5D utility before revision surgery | 0.120 | 0.363 | 0.544 | 0.054 | 0.355 | 0.474 |
| EQ-5D utility after revision surgery | 0.437 | 0.673 | 0.783 | 0.293 | 0.572 | 0.684 |
| B. Costs (in £) | | | | | | |
| Costs of primary arthroplasty | 5687 | 5502 | 5442 | 6158 | 6036 | 6021 |
| Costs of revision arthroplasty | 7768 | 7957 | 8160 | 7804 | 7751 | 7678 |
| Costs in the 12 months before arthroplasty | 579 | 416 | 320 | 963 | 898 | 513 |
| Readmissions<12 months after primary arthroplasty | 740 | 422 | 314 | 888 | 499 | 404 |
| Ambulatory costs<12 months after primary arthroplasty | 52 | 131 | 47 | 446 | 352 | 286 |
| Readmissions>12 months after primary arthroplasty | 161 | 115 | 85 | 266 | 169 | 120 |
| Readmissions in the year of revision surgery | 675 | 747 | 621 | 959 | 860 | 758 |
| Readmissions>12 months after revision surgery | 238 | 144 | 73 | 360 | 225 | 150 |
| Ambulatory costs>12 months after primary arthroplasty | – | – | – | 48 | 27 | 19 |
| Ambulatory costs in the year of revision surgery | – | – | – | 816 | 533 | 351 |
| Ambulatory costs for patients after revision surgery | – | – | – | 128 | 155 | 36 |

The columns show average predicted quality of life and costs for patients with different preoperative OHS and OKS scores. Predictions are averaged across the estimation sample. Columns 1 and 4 show values for patients at the 5th percentile of the OHS/OKS distribution. Columns 2 and 4 show outcomes for patients at the median of OHS/OKS distributions, and columns 3 and 6 show outcomes for patients at the 95th percentile of the OHS/OKS distribution.
EQ-5D, EuroQoL-5 Dimensions; THR, total hip replacement; TKR, total knee replacement.

Two previous studies on the cost-effectiveness of TKR and THR, respectively, found that while OHS/OKS were systematically associated with both costs and outcomes of TJR, both procedures were nonetheless cost-effective for almost all patients.[3 4] Similarly, our results indicate that quality of life improved for almost all patients undergoing TKR and THR. While the improvement was largest for patients with low scores, we also found that costs tended to be higher for patients with low scores.

A recent study concluded that, in the USA, access to TKR could be restricted to patients with more severe symptoms.[14] Similar to our findings, they reported that changes in quality of life were lower for patients with fewer preoperative symptoms (measured by SF-12 physical score). However, in their study TKR had minimal effects on quality of life on average (SF-6D utility increased by 0.008). Our findings contrast strongly with this, indicating that at least in the UK TJR is associated with sizable increases in EQ-5D-3L utility (average change of 0.32 for TKR and 0.45 for THR) for almost all patients currently undergoing surgery. This discrepancy may be due to differences between EQ-5D-3L and SF-6D, differences between our sample covering almost all UK knee replacement procedures and a small US sample of patients with early osteoarthritis, or methodological differences between the regression models selected for predictive accuracy in this study and marginal structural models used in the previous study.

**Table 3** Predicted quality of life and costs by age at operation

| Outcome | THR | | | TKR | | |
|---|---|---|---|---|---|---|
| | Age<60 | Age 60–79 | Age≥80 | Age<60 | Age 60–79 | Age≥80 |
| **A. Quality of life** | | | | | | |
| Postoperative EQ-5D utility | 0.778 | 0.794 | 0.729 | 0.637 | 0.731 | 0.722 |
| Preoperative EQ-5D utility | 0.316 | 0.348 | 0.281 | 0.310 | 0.404 | 0.397 |
| EQ-5D utility>12 months after primary arthroplasty | – | – | – | 0.616 | 0.689 | 0.645 |
| EQ-5D utility before revision surgery | 0.301 | 0.356 | 0.386 | 0.225 | 0.329 | 0.312 |
| EQ-5D utility after revision surgery | 0.572 | 0.670 | 0.562 | 0.418 | 0.558 | 0.521 |
| **B. Costs (in £)** | | | | | | |
| Costs of primary arthroplasty | 5420 | 5504 | 5763 | 5939 | 6039 | 6255 |
| Costs of revision arthroplasty | 8108 | 7938 | 7991 | 8020 | 7685 | 7390 |
| Costs in the 12 months before arthroplasty | 626 | 394 | 398 | 1150 | 804 | 634 |
| Readmissions<12 months after primary arthroplasty | 370 | 430 | 726 | 504 | 519 | 762 |
| Ambulatory costs<12 months after primary arthroplasty | 151 | 96 | 62 | 561 | 353 | 243 |
| Readmissions>12 months after primary arthroplasty | 135 | 117 | 110 | 218 | 171 | 156 |
| Readmissions in the year of revision surgery | 633 | 761 | 1344 | 763 | 857 | 941 |
| Readmissions>12 months after revision surgery | 159 | 151 | 139 | 428 | 242 | 105 |
| Ambulatory costs>12 months after primary arthroplasty | – | – | – | 50 | 27 | 27 |
| Ambulatory costs in the year of revision surgery | – | – | – | 711 | 556 | 160 |
| Ambulatory costs for patients after revision surgery | – | – | – | 145 | 124 | 118 |

The columns show average predicted quality of life and costs for patients in different age groups at primary operation. Predictions are averaged across the estimation sample.
EQ-5D, EuroQoL-5 Dimensions; THR, total hip replacement; TKR, total knee replacement.

Since OHS and OKS are already being used to determine eligibility for TJR,[6] evidence on their impact on the cost, efficacy and cost-effectiveness of primary arthroplasty is required to inform guidelines. The results presented in this study provide evidence on the association between preoperative scores and important measures of costs and quality of life. Moreover, the models presented here can be used as inputs in future model-based studies assessing the cost, efficacy and/or cost-effectiveness of THR and TKR, and in particular to explore heterogeneity between patient subgroups.

In this study, we used data from a range of different sources, including clinical trials and comprehensive administrative databases. The sample sizes in this study are up to 1000 times larger than those used in previous studies.[3 4 14] Crucially, this reduces the uncertainty around our estimates for patients with very high scores—that is, patients who are most likely to be affected by rationing. Our data also had a longer follow-up duration for TKR

than previous studies (up to 12 years compared with 5 years in previous studies[4 14]). Moreover, we employed a systematic model selection procedure to select the most appropriate statistical model from a range of different candidate models.

Nevertheless, there are several limitations to our study. KAT and COASt are markedly smaller than NHS PROMs-HES, and consequently models estimated on those datasets have more uncertainty and are relatively parsimonious. On the other hand, these datasets allowed us to estimate models for a larger range of relevant outcomes. While NHS PROMs-HES data include information on (nearly) all elective TJR procedures carried out in the NHS England, it nevertheless is a selected sample. In particular, preoperative scores are already being used in some regions to restrict access to TJR, which affects the distribution of preoperative scores in the data. Moreover, the findings of this study might not hold for patients that are currently denied access to TJR if those patients

differ systematically from those observed in the data. A report commissioned by the Royal College of Surgeons in 2014 noted that 16 out of 52 Clinical Commissioning Groups restricted access to THR based on an OHS threshold score.[6] However, current guidelines do not recommend a threshold for referral, and thus existing threshold scores are not based on evidence and they are not applied systematically. Therefore, these existing approaches should not have a major impact on our findings. We were not able to draw conclusions on total costs or cost-effectiveness based on the findings of this study. This requires long-term modelling work, for which the models presented in this study could serve as parameters. The costs derived in this study could be affected by coding and classification errors. For example, we identified primary arthroplasty episodes in NHS PROMs-HES data based on links between NHS PROMs questionnaires and HES episodes instead of narrowly defined procedure and diagnosis codes. We were also not able to distinguish between one-stage and two-stage revisions. Consequently, our estimates of the costs could be affected by miscoded diagnoses, procedures or data linkage. We used specific criteria to determine whether hospital readmissions were related to primary arthroplasty. Similarly, in COASt and KAT patients were asked to report health services used for problems related to their joint and their primary arthroplasty operation. Therefore, ambulatory and hospital readmission costs might be affected by classification errors. While coding and classification errors might bias the costs reported in this study, they are unlikely to affect the estimated associations between OHS/OKS and costs since it does not seem plausible that the frequency of such errors is related to patients' preoperative score. The costs were derived from the National Tariff as well as published reference costs. In some cases, these costs might not reflect the true costs to the hospital, although there is no reason to suspect that they would differ systematically.

We conducted a complete-case analysis since the use of multiple imputation would have further increased the complexity of our analysis. However, there is little reason to expect that missing data would be systematically related to the shape of the relationship between OHS/OKS and costs or quality of life. While we studied a wide range of outcome variables, we were not able to examine revision rates or long-term outcomes for THR due to a lack of appropriate data. We also focused on a limited range of covariates (OHS/OKS, age and sex). We did not investigate the impact of other covariates that may affect our outcomes and are used in clinical decision-making (eg, body mass index) due to a lack of data. While we tested a wide range of candidate models, our model selection was done sequentially, and therefore it is possible that there are interactions between model components that were not considered in this study. Our model selection process was developed to assess prediction accuracy, and therefore we did not consider statistical significance or confounding when selecting our models. Finally, our model selection process did not take into account whether differences

in model fit were statistically significant and it did not penalise model complexity.

These limitations affect our findings to varying degrees: There is very little uncertainty around the associations between OHS/OKS and quality of life as well as the cost of primary arthroplasty since the underlying models were estimated on large samples. Similarly, our findings for the cost of hospital readmissions were based on large samples; however, the cost variables are more likely to be affected by the coding and classification errors mentioned above. While our findings on ambulatory costs and cost after revision arthroplasty are based on the best available datasets, there is still considerable uncertainty around these estimates due to relatively small sample sizes as well as coding and classification errors. These shortcomings could be addressed in future research, for example, through the use of administrative datasets on primary care such as the Clinical Practice Research Datalink as well as data on revision arthroplasty from the National Joint Registry. Unfortunately, these datasets were not available for this research.

In summary, our results show that preoperative OHS and OKS are systematically associated with the costs and quality of life of patients undergoing TJR. However, further research is needed to estimate cost-effectiveness, and to determine the impact on revision rates and long-term outcomes for THR.

**Acknowledgements** The authors thank the ACHE investigators, user group and steering group for their role in the project. The authors also thank the KAT study group for providing access to their data for use in this project, as well as all of the researchers and participants involved in that study. They also thank all the participants of the COASt Study, Professor Nigel Arden and the COASt team for their time and dedication, and the NIHR for their funding support. Finally, they thank Chaudhry Shah and Joaquim Soares do Brito for their help with identifying the relevant codes for hospital readmissions used in the HES data.

**Contributors** AJP, DB, HAD, AMG and NKA conceived, designed and conducted the ACHE study. PE, HAD and AMG designed and conducted the regression analysis reported in this manuscript. PE and HAD analysed the data. PE drafted the manuscript. All authors edited the manuscript for important intellectual content and approved the final version. The authors had full access to the data (including statistical reports and tables), can take responsibility for the integrity of the data and the accuracy of the data analysis and are responsible for submitting this paper.

**Funding** All authors received a grant from NIHR funding the current work. KAT was funded by the NIHR as an HTA (No. 95/10/01). COASt was funded by the National Institute for Health Research (NIHR) under its Programme Grants for Applied Research Programme (reference number RP-PG-0407-10064). This study was conducted as part of the project 'Introducing Standardised And Evidence-Based Thresholds for Hip and Knee Replacement Surgery – The Arthroplasty Candidacy Help Engine (The ACHE Tool)', which was funded under the UK National Institute of Health Research (NIHR) Health Technologies Assessment (HTA) Programme (HTA 11/63/01). Visit the HTA programme website (www.hta.ac.uk) for further project information. The funder had no involvement in the collection, analysis and interpretation of data; in the writing of the report; and in the decision to submit the article for publication.

**Disclaimer** The views and opinions expressed in this paper are those of the authors and do not necessarily reflect those of the HTA programme, NIHR, NHS, NHS Digital or the Department of Health. The manuscript's guarantor (PE) affirms that the manuscript is an honest, accurate and transparent account of the study being reported; that no important aspects of the study have been omitted; and that any discrepancies from the study as planned (and, if relevant, registered) have been explained.

**Competing interests** HAD reports personal fees from Halyard Health, outside the submitted work. NKA reports grants from Bioiberica, personal fees from Bioventus, personal fees from ESCEO, personal fees from Flexion, personal fees from Freshfields Bruckhaus Deringer, personal fees from Merck, personal fees from Regeneron, outside the submitted work.

**Patient consent** Obtained.

**Ethics approval** No primary data was collected for this study, and therefore ethical approval was not required. The COASt study obtained ethical approval from Oxford research Ethics Committee (REC Ref: 10/H0604/91). The KAT trial was approved by the Multi Centre Research Ethics Committee for Scotland in November 1998 (research protocol MREC/98/0/100) and was approved by the Local Research Ethics Committees in each study centre recruiting trial participants.

**Provenance and peer review** Not commissioned; externally peer reviewed.

**Data sharing statement** Data on NHS PROMs linked to HES APC data were reused with the permission of NHS Digital, copyright 2015, with all rights reserved. The statistical code as well as variance–covariance matrices for the analyses reported in this study are available from the authors on request. PROMs/HES data are available from NHS Digital (http://content.digital.nhs.uk/dars); enquiries about access to the KAT data should be directed to kat@abdn.ac.uk; enquiries about access to the COASt data should be directed to coast.study@ndorms.ox.ac.uk

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
