## [Reviewer comments · BMJ Open]

ARTICLE DETAILS

TITLE (PROVISIONAL)	Association between pre-operative Oxford hip and knee scores and costs and quality-of-life of patients undergoing primary total joint replacement in the NHS England: An observational study
AUTHORS	Eibich, Peter; Dakin, Helen; Price, Andrew; Beard, David; Arden, Nigel; Gray, Alastair

VERSION 1 – REVIEW

REVIEWER	David Gwynne-Jones Department of Orthopaedic Surgery Dunedin School of Medicine, University of Otago, New Zealand
REVIEW RETURNED	20-Sep-2017

GENERAL COMMENTS	Abstract This is quite well written and summarises the key points of the paper. General There is a lot on the statistical methods and modelling used. Is this paper describing statistical modelling or is it attempting to answer the research question posed in the introduction. Or is it trying to refute the use of the OHS/OKS in rationing? What is the target audience? As an orthopaedic surgeon involved in service planning and delivery, I find the results as presented are unintelligible especially table 2 and various appendices. There is little analysis, clinical understanding or discussion of the results. I think this would improve the paper. Methods It is not clear if these are all elective procedures. Does it include THR for fracture or acute admission with end stage OA resulting in TJR. These patients are likely to have very low scores and much greater early costs than those admitted for elective surgery. Results Page 12 lines 28-40 I think this could be rewritten and clarified. The benefit in those with high OHS,OKS is marginal. No values are
--

given.

Lines 42 to 48 and table 2

This is unclear. Post op utilities are not included in table 2 which only gives coefficients. No values are given. Why was age 50-54 used as reference? Not mentioned in methods. This may be clear to a statistician but not to a general reader.

Page 14 Costs

These are greater with lower oxford scores below 10. Figure 1c appears dramatic but the scale on y axis does not go down to zero. The real difference is around £600 which is only 10% of total cost.

No values given for costs difference between male and female despite being stated to be statistically significant.

Costs of revisions

Similar comments apply as above. 10% difference in hip costs and nil for knees. No values given between male and females.

Costs in year before and after TJR

2A This result is self-evident. Some values would be helpful. It appears to be £200-500. The numbers used from the Coast study are so low that the 95% CIs become relatively meaningless especially in 2B.

The costs after TJR for hip are very low. How can they be negative? There is uncertainty (line 43) due to small numbers but this has not been mentioned with regard to 2A and B.

Decreased costs for knees with better preop function are intuitive.

Costs of readmissions increase with worse preop function as expected with more disabled patients. No data presented on costs for males vs females again.

Costs of readmission > 2 years after THR and TKR show a significant trend but the values are small (Change only £100 THR, £250 TKR)

Line 57 and p 15 line 3 to 4 mention costs in year 2 while graphs refer to > 2 years. Please clarify.

Page 15 line 9 should read Figure 3C and 3D

Lines 11-15 and fig 3 E,F Is this a relevant measure? Would preop primary OHS,OKS be expected to have a major impact on readmissions > 12 months after subsequent revision?

	Discussion Line 33-35 May incur greater costs. These could be quantified based on the results. Lines 37 to 41 This is important and could be discussed a bit further based on these results. There is some repetition in the first 3 paragraphs of discussion and this could be improved. Page 16 line 30 -33. This is the main weakness of the paper. Combining the 3 datasets may be valid but extrapolating from 400 to 200,000 makes the results questionable. Line 55 This may be overstated . There may be a strong statistical association but is this a clinically or financially relevant association? Some of the results presented could be discussed . Why could there be a difference between males and females, hips and knees, high and low scores.
--	---

REVIEWER	Bart Ferket Icahn School of Medicine at Mount Sinai, NY, USA
REVIEW RETURNED	23-Sep-2017

GENERAL COMMENTS	This is a comprehensive analysis using various osteoarthritis patient cohorts to address the association between preoperative Oxford hip and knee scores (OHS/OKS) vs. osteoarthritis and TJR related healthcare costs and preference-weighted EQ-5D quality-of-life index obtained pre-/post-TJR. The manuscript generally reads well, and provides a lot of detail on the statistical methods. The conclusions are interesting and important: EQ-5D utility improves in all patients with OHS (OKS) scores ≤ 46 (44), albeit that improvements were smaller in those with pre-TJR scores above 10, and “condition-related” healthcare costs are lower for those with lower scores. I have a few comments for the authors that may help improving the manuscript. Major comments: 1) Methods: Was it not possible to also analyse ambulatory and hospital readmission costs in the year after primary arthroplasty based on data from COAST? This would be an interesting analysis: comparing 1-yr costs accumulated before vs 1-yr costs accumulated after TJR. 2) Methods: Why did the authors limit healthcare costs to presumably procedure/condition-related costs only? Why not estimate presumably related and unrelated costs? It can be very difficult to disentangle related from unrelated costs, and it is generally recommended to also use total costs.
--

	3) Results: The Results section starts with showing the estimated model coefficients for associations (Table 2), accompanied with illustrations by Figures 1-2. Would it be possible to start with a simple description of the study population characteristics and the “observed” outcomes before and after TKR: not based on modelling? 4) Results: The figures would be most informative if they would represent adjusted effects of the OHS/OKS by keeping all other covariates constant at a specified value (e.g. median age) or just using the whole sample distribution for fixed OHS/OKS while generating predictions. The latter seems to be the case when I read the legends, but I cannot find it in the Methods section. 5) Discussion: In the Methods section it is stated that the analysis focused on predictive accuracy rather than causal inference. This statement seems in agreement with the way the outcome variables were selected: from pre- and post-operative measurements. It has been shown that regression-to-the mean is an important phenomenon in the context of TJR, see for example PMID 28127856. However, the Discussion section tends to assume causality: “delaying TJR surgery until patient’s symptoms deteriorate may not be appropriate”. It seems the authors assume that the improvements seen can be attributed to TJR, a conclusion that cannot be made based on their findings I believe. Also the change in costs with TJR needs to be analysed in order to make such a statement. Minor comments: 1) Abstract: Would it be possible to show numbers in the Results? 2) Discussion: “We also found ... above 10 or 20”. “also” should probably be replaced by “particularly” or something similar.
--	---

REVIEWER	Mark Pennington King's college London UK
REVIEW RETURNED	25-Sep-2017

GENERAL COMMENTS	The paper sets out to assess the impact of pre-operative patient characteristics (primarily OHS and OKS) on costs and quality of life in patients undergoing THR and TKR. Overall, I thought this was a thorough and clearly presented analysis which provides useful a resource for those interested in the cost-effectiveness of THR and TKR. In particular, I thought the approach to fitting regression models was robust and objective. I did wonder if there were sometimes much simpler model specifications that would have proved almost as good with regard to explanatory power and might have been preferred with regard to parsimony. In the respect I wasn't sure if the author's chosen measure of model fit MSE penalized models for addition terms (I think not?) and I would have preferred the use of a measure that did. However, this is a minor issue. The main results of the analysis are provided in a series of tables which I found clear and appropriate. However, I was left longing for some summary measures of the impact of patient characteristics on
---

	each of the dependent variables. I would have liked to see a summary table or tables reporting the predicted values for the dependent variables across a range of values for OKS/OHS and age, perhaps at the median and IQR for the independent variables? This could be based on predicted values with the other variables at their mean values? It would be nice to have some simple tabulation of the impact of age and OKS/OHS on the dependent variables to assess the extent of the impact. I appreciate that this information is conveyed in the figures and can be extracted. But I do think an additional table or two would be justified. I think it's worth noting (perhaps as a footnote) that NHS PROMs outcome data can be obtained up to 12 months after surgery. I'm not sure what proportion of the data is collected after six months, but unless it's really rather small I think it should probably be noted. The risk of a coding error resulting in a second ipsilateral primary joint replacement being wrongly classified as a revision and linked to the first primary joint replacement should probably be acknowledged. You comment that you could not reliably identify the type of operation from the OPCS codes. I agree that this would be subject to error. If I have understood correctly your costs were based on national tariffs for elective surgery. Did you apply the adjustments for length of stay beyond the trim points? Aside from this, (and assuming that you did not apply local Market Forces Factor corrections) the costs for hip surgery will have had one of two values according to whether the OPCS code and resulting payment was for a cemented or cementless procedure? What is the reason for the shape fall in primary costs with increasing OHS/OKS at low values? Is this simply due to excess bed stay in these patients? I think the manuscript would benefit from a little more clarification of the extent of variation of the resulting costs in the data, and the extent to which the costs (and particularly differences in costs) were representative of the true costs to hospitals. I wasn't completely clear whether the costs of revision arthroplasty were regressed on OHS/OKS scores before the original primary surgery or before the revision procedure. It might be helpful to clarify this. The costs of revision are likely to be highly dependent upon whether the revision is undertaken for infection and a two stage procedure is undertaken or the revision is one stage. I was not clear to what extent you were able to distinguish two stage revisions (from readmissions or further revisions etc)? To what extent did you data differentiate different revision costs? Are increasing costs with pre-operative OHS score a reflection of a greater proportion of infections and hence of two stage revisions in patients scoring higher?
--	--

REVIEWER	Nicholas Clement Royal Infirmary of Edinburgh UK
REVIEW RETURNED	02-Oct-2017

GENERAL COMMENTS	The authors should be congratulated on their work. This is an excellent paper. It is written well and conveys a very complex piece of work in simple terms for the lay reader to understand. The content adds significantly to the literature. I have a reasonable knowledge of statistics and all looks good with this paper, but it may benefit from a statistical review.
--

REVIEWER	Julie Agel
-----------------	------------

	Harborview Medical Center U.S.A.
REVIEW RETURNED	06-Nov-2017

GENERAL COMMENTS	This is an important manuscript that addresses a potentially important question. My paraphrase of that question would be who would benefit via improved quality of life and/or financial feasibility from a primary total knee or total hip or what is the benefit based on a given outcome score. The authors have data to address this question but the answer is buried in a great deal of data from multiple sources that is difficult to follow. I think this manuscript suffers from having too much data available that detracted from the primary message. I would consider building a consort diagram that allows the reader to track from your surgical sample what pre and post operative data you have including readmissions. Please explain why you chose to include information past one year if that is the limit of your cost data; especially revisions after one year. I would consider presenting one 'paper' on cost and another on quality of life for a cleaner presentation. This would make understanding your sample sizes clearer. You have to censure outcomes such as revision to the shortest time period of follow-up unless all that information came from the hospital data set. When are the questionnaires administered in relationship to surgery -- day of -- or time of surgical scheduling - how are the follow-up questionnaires obtained My biggest source of confusion is what you are using as an independent variable - did you use everything in Table 1? Did you consider using change within a person on the EQ-5D. are all the Oxford hip and knee scores pre-primary total joint only? Since this is complicated statistics perhaps Appendix D should include an example. Perhaps change your abstract objective to specify that quality of life is defined as the EQ-5D. How are OHS and OKS being used to determine eligibility now - how do those cut-offs fit in your models; does the fact that these cutoffs are apparently in place in some settings impact your NHS data
--

VERSION 1 – AUTHOR RESPONSE

We thank you very much for your constructive comments and suggestions. The constructive critique helped us to improve the quality of the manuscript significantly. We believe that we were able to address all concerns and questions raised. Please find our responses to your points below.

RESPONSES TO COMMENTS BY THE EDITOR

Comment:

- Please include the study design and setting in the title. This is the preferred format of the journal.
- Please include an ethical approval statement in your manuscript, explaining why you do not have/need ethical approval.
- Please include a clear 'Methods' heading before you describe the data set.

Response:

We apologise for this oversight. We have made the requested changes in the revised version of the manuscript.

RESPONSES TO COMMENTS BY REVIEWER #1

Comment:

Ethics not discussed in the paper.

Response:

We have added a statement on ethical approval at the end of the manuscript (p.20).

Comment:

Abstract

This is quite well written and summarises the key points of the paper.

General

There is a lot on the statistical methods and modelling used. Is this paper describing statistical modelling or is it attempting to answer the research question posed in the introduction. Or is it trying to refute the use of the OHS/OKS in rationing?

What is the target audience? As an orthopaedic surgeon involved in service planning and delivery, I find the results as presented are unintelligible especially table 2 and various appendices. There is little analysis, clinical understanding or discussion of the results. I think this would improve the paper.

Response:

Thank you very much for your comment. This study had two aims: On the one hand, we examined the relationship between OHS/OKS and quality-of-life as well as costs of patients undergoing TJR. This involves statistical modelling, but nevertheless we believe that the results are primarily of interest to clinicians and policy makers, since OHS/OKS are already being used in rationing. In addition, we think that the statistical modelling and the regression results are interesting in their own right, since they could be used as input parameters for future modelling studies (e.g., health economic modelling) in this area. The latter aim is the reason for presenting detailed modelling results in the appendices. However, we accept that the presentation of some of these results (such as Table 2 in the original manuscript) detracted from the message of the paper. Therefore, we have made several changes to the manuscript. We have moved Table 2 from the main text into Appendix C, which now contains all statistical models. We have added two new tables to the manuscript (Table 2 and 3 in the revised version), which summarize the predicted quality-of-life and costs for patients with different OHS/OKS scores and different ages. We have also rewritten the results and discussion section to further discuss these differences.

Comment:

Methods

It is not clear if these are all elective procedures. Does it include THR for fracture or acute admission with end stage OA resulting in TJR. These patients are likely to have very low scores and much greater early costs than those admitted for elective surgery.

Response:

We only included patients undergoing elective primary arthroplasty. We have added a sentence on p. 4 in the revised version of the manuscript to clarify this.

Comment:

Results

Page 12 lines 28-40 I think this could be rewritten and clarified. The benefit in those with high OHS,OKS is marginal. No values are given.

Response:

We have rewritten the paragraph and added utility values for certain OHS/OKS scores. In the revised version, the paragraph now reads as follows (p.12):

“In contrast, post-operative quality-of-life increased strongly from 0.39/0.24 for OHS/OKS of 0 to 0.72/0.61 for scores of 10, after which there were only small differences in quality-of-life by OHS/OKS. As a consequence, the health gain associated with TJR (i.e., the difference between the two curves) was largest for patients with low scores (0.69 for OHS of 5 and 0.53 for OKS of 6). While the models predicted that post-operative EQ-5D utility was higher than pre-operative utility for all patients with OHS \leq 46 and those with OKS \leq 44, the differences between post- and pre-operative utility were marginal (0.005 for OHS of 46 and 0.02 for OKS of 44).”

Comment:

Lines 42 to 48 and table 2

This is unclear. Post op utilities are not included in table 2 which only gives coefficients. No values are given. Why was age 50-54 used as reference? Not mentioned in methods. This may be clear to a statistician but not to a general reader.

Response:

Figure 1A-B is based on the regression results in Table 2. In order to include binary indicators (in this case for age) in the regression, we had to choose one category as the reference group. However, the choice of the reference group does not affect the results qualitatively. In the revised manuscript we have moved Table 2 into Appendix C, and therefore we have cut the corresponding paragraph from the main text.

Comment:

Page 14 Costs

These are greater with lower oxford scores below 10. Figure 1c appears dramatic but the scale on y axis does not go down to zero. The real difference is around £600 which is only 10% of total cost. No values given for costs difference between male and female despite being stated to be statistically significant.

Costs of revisions

Similar comments apply as above. 10% difference in hip costs and nil for knees. No values given between male and females.

Response:

We have added the values corresponding to the OHS/OKS scores mentioned in the paragraph. We have also added average values for the cost differences between men and women and corrected a mistake in the interpretation. From these figures (as well as Table 2 in the revised manuscript) it should become clear that the difference in cost by OHS/OKS is relatively modest. We also explicitly acknowledge this in the second paragraph of the discussion section (p. 17):

“While the differences in costs are moderate compared to differences in quality-of-life (between £100 and £500 for most outcomes), it is worth noting that some of these costs are incurred annually, and therefore even moderate differences can be economically important in the long-run.”

Comment:

Costs in year before and after TJR

2A This result is self-evident. Some values would be helpful. It appears to be £200-500.

Response:

We have added the corresponding values into the text.

Comment:

The numbers used from the Coast study are so low that the 95% CIs become relatively meaningless especially in 2B. The costs after TJR for hip are very low. How can they be negative? There is uncertainty (line 43) due to small numbers but this has not been mentioned with regard to 2A and B.

Response:

The negative predicted costs in Figure 2C are an artefact of the statistical model. The linear regression model used for Figure 2C does not restrict the outcome to be non-negative, and therefore predictions from this model can be negative although the observed costs are not. We have added a sentence to the manuscript to explain this. We have also revised the sentence in line 43 in the original manuscript to state that Figures 2A-D are all based on substantially smaller numbers than the preceding figures.

Comment:

Decreased costs for knees with better preop function are intuitive.
Costs of readmissions increase with worse preop function as expected with more disabled patients.
No data presented on costs for males vs females again.

Response:

We have added values for average differences between men and women to the paragraph.

Comment:

Costs of readmission > 2 years after THR and TKR show a significant trend but the values are small (Change only £100 THR, £250 TKR)

Response:

We have added the corresponding values to the text.

Comment:

Line 57 and p 15 line 3 to 4 mention costs in year 2 while graphs refer to > 2 years. Please clarify.

Response:

We have revised the wording of the relevant outcome variables to better reflect that these are annual readmission cost between 1 and 6 years after primary arthroplasty.

Comment:

Page 15 line 9 should read Figure 3C and 3D

Response:

We apologise for this mistake, we have corrected it in the revised version.

Comment:

Lines 11-15 and fig 3 E,F Is this a relevant measure? Would preop primary OHS,OKS be expected to have a major impact on readmissions > 12 months after subsequent revision?

Response:

While it is possible that pre-operative pre-primary OHS/OKS is related to readmissions after revision surgery, we agree that it is not likely to have a major impact. The estimation results confirm this. However, the aim of this study was to provide comprehensive evidence on the relationship between cost and OHS/OKS, and therefore we included this measure for the sake of completeness.

Comment:

Line 33-35 May incur greater costs. These could be quantified based on the results.

Lines 37 to 41 This is important and could be discussed a bit further based on these results.

There is some repetition in the first 3 paragraphs of discussion and this could be improved.

Response:

We have rewritten the first three paragraphs of the discussion section to better reflect the findings. Table 2 and 3 at the end of the results section also provide values for costs and quality-of-life of patients with different OHS/OKS scores and ages. The differences between patients with very low and very high scores are discussed in the corresponding paragraphs on pp. 13/14.

Comment:

Page 16 line 30 -33. This is the main weakness of the paper. Combining the 3 datasets may be valid but extrapolating from 400 to 200,000 makes the results questionable.

Response:

We agree that this is a weakness, but thought it important for completeness to include estimates for ambulatory costs, costs before surgery and long-term changes in quality-of-life, which unfortunately were not available from PROMs-HES data. Nevertheless, we acknowledge in the limitations that there is more uncertainty around the estimates from KAT and COAST.

Comment:

Line 55 This may be overstated. There may be a strong statistical association but is this a clinically or financially relevant association?

Response:

We argue that the associations with quality-of-life are large and clinically relevant, and while the associations with costs might be more moderate, it is likely that these differences are relevant in the long-run, since some of the costs (e.g., for hospital readmissions) refer to annual costs. This is discussed in the first two paragraphs of the discussion section in the revised version of the manuscript (p.16/17):

“OHS and OKS were developed as outcome measures reported by patients undergoing THR and TKR. Previous studies found that OHS and OKS are correlated with both clinical scores (such as the Charnley score or the American Knee Society score) and patient-reported general health measures (such as SF-36).[27,28] Thus, it is not surprising that we found that higher scores were systematically associated with higher quality-of-life. However, the difference between post-operative and pre-operative EQ-5D utility decreased with OHS/OKS. Patients with low scores have worse symptoms and more functional limitations due to osteoarthritis. Therefore, they have more potential to benefit from primary arthroplasty.

We also found that costs are moderately higher for patients with low pre-operative scores. Patients with worse symptoms due to osteoarthritis might require longer recovery times. In addition, they are likely to have comorbidities or complications, which also increase the costs of primary arthroplasty.

While the differences in costs are moderate compared to differences in quality-of-life (between £100 and £500 for most outcomes), it is worth noting that some of these costs are incurred annually, and therefore even moderate differences can be economically important in the long-run.”

In the sentence mentioned in your comment, we have replaced the word “strongly” with “systematically” to highlight that we found statistically significant associations for almost all outcomes.

Comment:

Some of the results presented could be discussed . Why could there be a difference between males and females, hips and knees, high and low scores.

Response:

We have rewritten the first two paragraphs of the discussion section to include a short discussion of possible causes for the estimated differences between patients with low and high scores. The differences between men and women are relatively minor (as quantified in the revised version).

RESPONSES TO COMMENTS BY REVIEWER #2

Comment:

This is a comprehensive analysis using various osteoarthritis patient cohorts to address the association between preoperative Oxford hip and knee scores (OHS/OKS) vs. osteoarthritis and TJR related healthcare costs and preference-weighted EQ-5D quality-of-life index obtained pre-/post-TJR. The manuscript generally reads well, and provides a lot of detail on the statistical methods. The conclusions are interesting and important: EQ-5D utility improves in all patients with OHS (OKS) scores ≤ 46 (44), albeit that improvements were smaller in those with pre-TJR scores above 10, and “condition-related” healthcare costs are lower for those with lower scores. I have a few comments for the authors that may help improving the manuscript.

Major comments:

1) Methods: Was it not possible to also analyse ambulatory and hospital readmission costs in the year after primary arthroplasty based on data from COAST? This would be an interesting analysis: comparing 1-yr costs accumulated before vs 1-yr costs accumulated after TJR.

Response:

Thank you very much for the suggestion. We agree that this is an interesting question. In an earlier version of the paper we did use the COAST data to estimate ambulatory and hospital readmissions costs in the first 12 months after primary hip replacement. The results indicated that costs in the first year after arthroplasty are considerably lower than in the year before arthroplasty. For example, patients with an OHS of 17 (median value) had average costs of £416 in the first 12 months before hip replacement, and average costs of £221 in the first 12 months after hip replacement. However, we decided against adding these results into the revised version of the paper for two reasons. First, we think that the estimates of hospital readmission costs based on PROMs-HES data are more credible, since the sample size is considerably larger and the costing of readmission episodes is more precise, taking into account the procedures and diagnosis for each case. In contrast, the costs of hospital readmissions in COAST were derived using a weighted average of the reference costs for orthopaedic bed days. This difference is reflected in the results: while the average costs of hospital readmissions in the first 12 months after primary THR in COAST is £160, the average costs in PROMs-HES is £455. Second, some of the other reviewers felt that the paper already presents too many results, and therefore we were reluctant to add new analyses into the study.

Comment:

2) Methods: Why did the authors limit healthcare costs to presumably procedure/condition-related costs only? Why not estimate presumably related and unrelated costs? It can be very difficult to disentangle related from unrelated costs, and it is generally recommended to also use total costs.

Response:

It is certainly true that in some disease areas it can be difficult to determine a priori whether costs are related to the procedure/condition under study or whether they are unrelated. However, using total costs in this study would likely reduce the statistical precision of some of our models considerably, since the majority of, e.g., ambulatory costs in this age group are unlikely to be related to the procedure. This would have posed a problem for some of the models estimated on smaller samples. At the same time, we are confident that we were able to establish with reasonable accuracy whether certain costs were related to primary arthroplasty or not. Our classification was based on clinical expertise (AP, NA, DB) and extensive discussions with a clinical coder and a clinical registrar. Nevertheless, we acknowledge that classification errors are a potential source of bias. In the revised manuscript, we have added a sentence on p. 18 to acknowledge this:

“The costs derived in this study could be affected by coding and classification errors. For example, we identified primary arthroplasty episodes in NHS PROMs-HES data based on links between NHS PROMs questionnaires and HES episodes instead of narrowly defined procedure and diagnosis codes. We were also not able to distinguish between one-stage and two-stage revisions. Consequently, our estimates of the costs could be affected by miscoded diagnoses, procedures or data linkage. We used specific criteria to determine whether hospital readmissions were related to primary arthroplasty. Similarly, in COASt and KAT patients were asked to report health services used for problems related to their joint and their primary arthroplasty operation. Therefore, ambulatory and hospital readmission costs might be affected by classification errors. While coding and classification errors might bias the costs reported in this study, they are unlikely to affect the estimated associations between OHS/OKS and costs, since it does not seem plausible that the frequency of such errors is related to patient’s pre-operative score.”

Comment:

3) Results: The Results section starts with showing the estimated model coefficients for associations (Table 2), accompanied with illustrations by Figures 1-2. Would it be possible to start with a simple description of the study population characteristics and the “observed” outcomes before and after TKR: not based on modelling?

Response:

Thank you very much for this suggestion. We have amended Table 1 to show the mean values and standard deviations for each quality-of-life or cost variable along with the mean OHS/OKS score in the respective sample. In addition, we discuss these figures on p.8 in the revised manuscript:

“Table 1 below provides an overview over all outcomes for which regression models were developed. Average post-operative EQ-5D utility was considerably higher than average pre-operative utility. Interestingly, average EQ-5D utility before revision surgery was similar to or even lower than average pre-operative utility. Average utility after revision was higher than average utility values before revision, however, it was lower than the average post-operative utility. On average, primary hip arthroplasty costed £5,522 and primary knee arthroplasty costed £6,053. Average cost in the 12 months before arthroplasty were £444 for THR and £836 for TKR. In the first 12 months after THR, average costs were £101 for ambulatory services and £455 for hospital readmissions. For TKR, average costs in the first 12 months after surgery were £361 for ambulatory services and £550 for hospital readmissions. Average pre-operative OHS scores varied between 16.8 and 18.7 (due to the different samples), while average pre-operative OKS scores varied between 16.9 and 19.4. For the

sake of brevity, we discuss the modelling results for selected outcomes in this paper; the models for all other outcomes are provided in the appendix.”

Comment:

4) Results: The figures would be most informative if they would represent adjusted effects of the OHS/OKS by keeping all other covariates constant at a specified value (e.g. median age) or just using the whole sample distribution for fixed OHS/OKS while generating predictions. The latter seems to be the case when I read the legends, but I cannot find it in the Methods section.

Response:

The figures are obtained by generating predictions for each observation in the sample, and then averaging across all observations with the same OHS/OKS score, which implies that the estimated association between OHS/OKS and costs/quality-of-life is averaged across the sample distribution of all other covariates. We have added the following sentence into the Methods section on p.12 to clarify this:

“We visualise the estimated associations between OHS/OKS and quality-of-life as well as costs by plotting predicted costs and quality-of-life against OHS/OKS scores. We generated predictions using the estimation sample, i.e., for each OHS/OKS score we averaged predictions over the sample distributions of the other covariates. The full estimation results as well as the candidate models considered for each outcome are provided in section C of the supplementary file.”

Comment:

5) Discussion: In the Methods section it is stated that the analysis focused on predictive accuracy rather than causal inference. This statement seems in agreement with the way the outcome variables were selected: from pre- and post-operative measurements. It has been shown that regression-to-the mean is an important phenomenon in the context of TJR, see for example PMID 28127856. However, the Discussion section tends to assume causality: “delaying TJR surgery until patient’s symptoms deteriorate may not be appropriate”. It seems the authors assume that the improvements seen can be attributed to TJR, a conclusion that cannot be made based on their findings I believe. Also the change in costs with TJR needs to be analysed in order to make such a statement.

Response:

It is correct that the focus of the analysis was on predictive accuracy rather than causality, and therefore we cannot interpret our findings as causal relationships without further assumptions. We accept that the relevant paragraph in the discussion section was not fully supported by our findings. While our findings suggest that pre-operative OHS/OKS scores could affect the cost-effectiveness of primary arthroplasty, we cannot draw any conclusions about this based on the findings in this study. In the revised version of the manuscript, we have rewritten the relevant paragraph on p. 17 to reflect this:

“Our findings do not allow us to draw conclusions about the cost-effectiveness of TJR for patients with different pre-operative scores without further assumptions about the costs and quality-of-life for patients without TJR. Nevertheless, based on these findings it appears promising to investigate differences in cost-effectiveness as well as the impact of rationing policies. In particular, the finding that the associations between pre-operative score and costs as well as quality-of-life are nonlinear suggests that there could be a threshold score beyond which TJR might not be considered beneficial or cost-effective. However, identifying this threshold is beyond the scope of this paper and left to future research.”

Comment:

1) Abstract: Would it be possible to show numbers in the Results?

Response:

We have changed the abstract to include some of the figures shown in the new summary Tables 2 and 3.

Comment:

2) Discussion: "We also found ... above 10 or 20". "also" should probably be replaced by "particularly" or something similar.

Response:

We have rewritten the relevant paragraph, please see our response to your comment #5.

RESPONSES TO COMMENTS BY REVIEWER #3

Comment:

The paper sets out to assess the impact of pre-operative patient characteristics (primarily OHS and OKS) on costs and quality of life in patients undergoing THR and TKR. Overall, I thought this was a thorough and clearly presented analysis which provides useful a resource for those interested in the cost-effectiveness of THR and TKR. In particular, I thought the approach to fitting regression models was robust and objective. I did wonder if there were sometimes much simpler model specifications that would have proved almost as good with regard to explanatory power and might have been preferred with regard to parsimony. In the respect I wasn't sure if the author's chosen measure of model fit MSE penalized models for addition terms (I think not?) and I would have preferred the use of a measure that did. However, this is a minor issue.

Response:

Thank you very much for your comment. Your observation is entirely correct – it is indeed possible that in some instances a simpler model might have yielded a similar model fit. The model selection criterion used in this study (minimal MSE) did not take model complexity into account. We considered alternative criteria (such as AIC) that penalize complexity. However, due to the scaling it is not straight forward to compare AIC values for models from different model classes (e.g., a linear regression model to a gamma-GLM). Therefore, we decided to use minimal MSE as our model selection criterion. Nevertheless, the choice of this measure has certain disadvantages. We have added a sentence to acknowledge these limitations in the revised version of the manuscript (p. 18):

"Finally, our model selection process did not take into account whether differences in model fit were statistically significant and it did not penalize model complexity."

Comment:

The main results of the analysis are provided in a series of tables which I found clear and appropriate. However, I was left longing for some summary measures of the impact of patient characteristics on each of the dependent variables. I would have liked to see a summary table or tables reporting the predicted values for the dependent variables across a range of values for OKS/OHS and age, perhaps at the median and IQR for the independent variables? This could be based on predicted values with the other variables at their mean values? It would be nice to have some simple tabulation of the impact of age and OKS/OHS on the dependent variables to assess the extent of the impact. I appreciate that this information is conveyed in the figures and can be extracted. But I do think an additional table or two would be justified.

Response:

Thank you very much for your suggestion. We have added two new tables in the revised version of the manuscript, which summarize the predicted outcomes for all measures at selected OHS/OKS

scores and ages (Table 2 and 3 in the revised manuscript). To accommodate these additions, we have also moved some material out of the main text into the Appendix.

Comment:

I think it's worth noting (perhaps as a footnote) that NHS PROMs outcome data can be obtained up to 12 months after surgery. I'm not sure what proportion of the data is collected after six months, but unless it's really rather small I think it should probably be noted.

Response:

In the revised version of the manuscript we have clarified that post-operative PROMs data is collected at minimum six months after primary arthroplasty (p.5).

Comment:

The risk of a coding error resulting in a second ipsilateral primary joint replacement being wrongly classified as a revision and linked to the first primary joint replacement should probably be acknowledged. You comment that you could not reliably identify the type of operation from the OPCS codes. I agree that this would be subject to error.

Response:

Thank you very much for pointing this out. This is a limitation of our study in general: e.g., it was also not possible for us to distinguish one-stage and two-stage revisions in the data, and it is entirely possible that the second-stage of a two-stage revision would be classified as a readmission instead. We have added the following paragraph in the revised version of the paper to acknowledge and discuss these limitations (p.18):

"The costs derived in this study could be affected by coding and classification errors. For example, we identified primary arthroplasty episodes in NHS PROMs-HES data based on links between NHS PROMs questionnaires and HES episodes instead of narrowly defined procedure and diagnosis codes. We were also not able to distinguish between one-stage and two-stage revisions.

Consequently, our estimates of the costs could be affected by miscoded diagnoses, procedures or data linkage. We used specific criteria to determine whether hospital readmissions were related to primary arthroplasty. Similarly, in COASt and KAT patients were asked to report health services used for problems related to their joint and their primary arthroplasty operation. Therefore, ambulatory and hospital readmission costs might be affected by classification errors. While coding and classification errors might bias the costs reported in this study, they are unlikely to affect the estimated associations between OHS/OKS and costs, since it does not seem plausible that the frequency of such errors is related to patient's pre-operative score."

Comment:

If I have understood correctly your costs were based on national tariffs for elective surgery. Did you apply the adjustments for length of stay beyond the trim points? Aside from this, (and assuming that you did not apply local Market Forces Factor corrections) the costs for hip surgery will have had one of two values according to whether the OPCS code and resulting payment was for a cemented or cementless procedure?

Response:

The costs for hospital admissions (including primary arthroplasty and revision episodes) include the cost of the base HRG as well as costs for excess bed days beyond the trim point and any unbundled costs, e.g., for diagnostic imaging. The costs take on a range of different values, this is (to the best of our knowledge) partly due to factors such as complications or comorbidities. We did not apply local Market Forces Factor corrections.

Comment:

What is the reason for the shape fall in primary costs with increasing OHS/OKS at low values? Is this simply due to excess bed stay in these patients?

Response:

We did look into this, and it appears that patients with low OHS/OKS values have higher costs for excess bed days, unbundled procedures as well as base HRG costs. The difference in base HRG costs is more pronounced than the difference in excess bed days and unbundled procedures, which suggests that patients with low OHS/OKS values are more likely to have complications or comorbidities.

Comment:

I think the manuscript would benefit from a little more clarification of the extent of variation of the resulting costs in the data, and the extent to which the costs (and particularly differences in costs) were representative of the true costs to hospitals.

Response:

We have amended Table 1 to show means and standard deviation of all costs and quality-of-life variables. We have also added the following sentence into the discussion section on p. 18:

“The costs were derived from the National Tariff as well as published reference costs. In some cases these costs might not reflect the true costs to the hospital, although there is no reason to suspect that they would differ systematically.”

Comment:

I wasn't completely clear whether the costs of revision arthroplasty were regressed on OHS/OKS scores before the original primary surgery or before the revision procedure. It might be helpful to clarify this.

Response:

All OHS/OKS scores referred to in this study are pre-operative pre-primary scores. We have added a sentence on p. 8 to clarify this.

“We used OHS/OKS scores measured before primary arthroplasty in all models.”

Comment:

The costs of revision are likely to be highly dependent upon whether the revision is undertaken for infection and a two stage procedure is undertaken or the revision is one stage. I was not clear to what extent you were able to distinguish two stage revisions (from readmissions or further revisions etc)? To what extent did your data differentiate different revision costs? Are increasing costs with pre-operative OHS score a reflection of a greater proportion of infections and hence of two stage revisions in patients scoring higher?

Response:

As mentioned above, we are unfortunately not able to distinguish one-stage and two-stage revisions, and it is possible that the second stage of a two-stage revision might be misclassified as a further revision or a readmission. We have acknowledged this in the limitations on p. 18:

“We were also not able to distinguish between one-stage and two-stage revisions. Consequently, our estimates of the costs could be affected by miscoded diagnoses, procedures or data linkage.”

Unfortunately, we cannot offer an answer as to why revision costs are increasing with pre-operative OHS. The HRG codes are fairly generic, and therefore it would require an in-depth review of diagnosis and procedure codes to answer this question.

RESPONSES TO COMMENTS BY REVIEWER #4

Thank you very much for your comment. We are glad to hear that you found the paper to be a significant contribution to the literature, and we are also glad to hear that the paper was easy to understand.

RESPONSES TO COMMENTS BY REVIEWER #5

Comment:

This is an important manuscript that addresses a potentially important question. My paraphrase of that question would be who would benefit via improved quality of life and/or financial feasibility from a primary total knee or total hip or what is the benefit based on a given outcome score.

The authors have data to address this question but the answer is buried in a great deal of data from multiple sources that is difficult to follow.

I think this manuscript suffers from having too much data available that detracted from the primary message.

Response:

Thank you very much for your comments. This study certainly was motivated by the question “which patients would benefit from total hip and total knee replacement?”, and while we cannot provide a definite answer we hope that the data presented in this study can provide important insights into this topic.

We accept that the study presents a lot of (and perhaps too much) data. However, we are reluctant to completely remove material from the study, since we think that the data presented in the manuscript might prove useful for future studies in this area (e.g., modelling studies of cost-effectiveness).

Moreover, we would prefer to present all the data together to facilitate use in future studies.

Nevertheless, we have made several changes to the revised manuscript, which hopefully should make the message clearer.

For example, we have moved Table 2 in the previous version (showing the regression model for post-operative EQ-5D) into the appendix, which now contains all tables of regression coefficients. We have also added two new tables (Table 2 and 3 in the revised version), partly in response to other referee comments, which provide a summary of average quality-of-life and costs for patients with different pre-operative scores as well as different ages.

Comment:

I would consider building a consort diagram that allows the reader to track from your surgical sample what pre and post operative data you have including readmissions.

Response:

Thank you very much for this suggestion. We carefully considered the use of a diagram to present the available data, but eventually decided against it as it would be very difficult to convey the necessary information about the different datasets in a concise figure. Instead, we have restructured and slightly expanded Table 1. In the revised version, Table 1 consist of two separate panels for quality-of-life

outcomes and costs outcomes, and we have also added columns showing the mean values and standard deviations for each outcome and the average pre-operative score in each sample.

Comment:

Please explain why you chose to include information past one year if that is the limit of your cost data; especially revisions after one year.

Response:

We apologise for this misunderstanding. While the data from the COASSt study is limited to a one-year follow-up, the other datasets used in this study have substantially longer follow-up durations (six years in NHS-PROMs and 12 years in KAT). Therefore, we included information past one year only for those outcomes where we have data available for a longer follow-up duration. For example, we included data on annual ambulatory costs between 1 and 12 years after total knee replacement, since such data was available from the KAT trial (see, e.g., the bottom three rows in Table 1). We did not include the corresponding figures for hip replacement, since the COASSt data was limited to one year of follow-up. Data on hospital readmissions and revisions was taken from PROMs-HES data with a follow-up of up to six years. We have adjusted the wording of some of our outcome variables to better reflect the follow-up duration.

Comment:

I would consider presenting one 'paper' on cost and another on quality of life for a cleaner presentation. This would make understanding your sample sizes clearer.

Response:

As outlined in our response to your first comment, we would prefer to present all the information in one study to avoid repetition of study objectives, methods and data sources, and to facilitate the use of these figures in future modelling studies. We have restructured Tables 1-3 into separate panels for quality-of-life and cost outcomes, which hopefully will make the presentation clearer.

Comment:

You have to censure outcomes such as revision to the shortest time period of follow-up unless all that information came from the hospital data set.

Response:

We apologise again for this misunderstanding. As described above, most datasets include substantially longer follow-up. Moreover, all cost outcomes are either procedure-related costs (e.g., costs of primary arthroplasty or costs of revision arthroplasty) or annual costs. We have clarified the wording of some of our outcome variables to reflect this.

Comment:

When are the questionnaires administered in relationship to surgery -- day of -- or time of surgical scheduling - how are the follow-up questionnaires obtained

Response:

The relevant guidance document by the Department of Health recommends that provider administer pre-operative questionnaires at the day of admission for surgery. However, this can be modified by the provider to better reflect local processes. In any case, the pre-operative questionnaire should be administered between the date when a patient is deemed fit for surgery and the date of surgery.

Follow-up questionnaires are sent out by post after a minimum period of six months after surgery. We have amended the relevant paragraph on p. 5.

Comment:

My biggest source of confusion is what you are using as an independent variable - did you use everything in Table 1?

Response:

The variables on costs and quality-of-life listed in Table 1 are used as dependent variables for our regression models. For most regression models we used pre-operative OHS or OKS, age at operation and sex of the patient as independent variables. In certain cases we considered additional variables. This is described in detail in the section "Covariates". In the revised version, we amended the figure captions to state the independent variables used in each model.

Comment:

Did you consider using change within a person on the EQ-5D.

Response:

We did consider using change in EQ-5D as an outcome variable. We ultimately decided against including such specifications in the model selection process for two reasons. First, we felt that it would further complicate the model selection process if we were to include specifications with different outcome variables. Second, the statistical distribution of EQ-5D utility is known and there are models to take this distribution (and in particular the lower and upper limit) into account. The possible change in EQ-5D, however, depends on baseline EQ-5D, and therefore we would have needed to adjust for baseline EQ-5D when analysing change in EQ-5D. This would have made model selection more difficult, which is why we decided against using change in EQ-5D as an outcome variable.

Comment:

are all the Oxford hip and knee scores pre-primary total joint only?

Response:

Yes, the OHS and OKS scores used in the study always refer to the scores measured before primary arthroplasty. We have added the following sentence on p. 8 to clarify this:

"We used OHS/OKS scores measured before primary arthroplasty in all models."

Comment:

Since this is complicated statistics perhaps Appendix D should include an example.

Response:

Thank you very much for this suggestion, we have added examples to all sections in appendix D.

Comment:

Perhaps change your abstract objective to specify that quality of life is defined as the EQ-5D.

Response:

We have revised the abstract accordingly.

Comment:

How are OHS and OKS being used to determine eligibility now - how do those cut-offs fit in your models; does the fact that these cutoffs are apparently in place in some settings impact your NHS data

Response:

A range of threshold values for OHS and OKS are being used in some regions to determine whether patients should be referred for surgical assessment. The scores might be used either in primary care or in so-called hubs (i.e., between primary and secondary care). Whether the scores are used to determine eligibility and how they are being used differs across regions, and unfortunately we do not have comprehensive data on this. We argue that due to the large sample size of the NHS PROMs - HES data it appears unlikely that these existing cut-offs affect the data substantially. However, we cannot rule out that our findings are affected by these cut-offs.

In the revised version of the manuscript, we rewrote the corresponding sentence in the introduction to clarify that these existing thresholds are only applied in certain regions. Moreover, we added a paragraph to the discussion section to acknowledge this limitation (p. 18):

“While NHS PROMs-HES data include information on (nearly) all elective TJR procedures carried out in the NHS England, it nevertheless is a selected sample. In particular, pre-operative scores are already being used in some regions to restrict access to TJR, which affects the distribution of pre-operative scores in the data. Moreover, the findings of this study might not hold for patients that are currently denied access to TJR if those patients differ systematically from those observed in the data. Unfortunately, it is not possible to determine this without further information on the triage process.”

VERSION 2 – REVIEW

REVIEWER	Mark Pennington King's Health Economics, King's College London, UK
REVIEW RETURNED	11-Dec-2017

GENERAL COMMENTS	I think the authors have thoroughly revised the paper and accommodated reviewer comments. The additional tables are helpful in my view. I noted one typo - the mean and standard deviation for Annual costs of readmissions in years 1-6 for THR was missing a minus. I also thought that the title of table 2 might benefit from the addition of 'at primary operation' or something similar just to highlight that the OHS/OKS data is always for the primary procedure and not prior to revision. I'll leave that to the author's discretion.
--

REVIEWER	David Gwynne-Jones Dept of Surgical Sciences Dunedin School of Medicine University of Otago Dunedin New Zealand
REVIEW RETURNED	13-Dec-2017

GENERAL COMMENTS	I think that the revised version is much more readable and the results and conclusions have been clarified.
---

REVIEWER	Julie Agel Harborview Medical Center USA
-----------------	---

REVIEW RETURNED	15-Dec-2017
-------------

GENERAL COMMENTS	This manuscript remains very dense reading although very thorough and thoughtful. I applaud the authors for not wanting to have multiple manuscripts but I think in this particular case two more separate manuscripts would help with the reading of the material; in particular one for clinicians and perhaps one for health economists. Part of the problem remains that you have a variety of places where you cannot draw conclusions due to the merging of data sets with different information and years. Your conclusion in the manuscript and the abstract should be the same. Based on all the variations of models etc a statement about your confidence in your findings might be helpful. If you were unable to prove the economics issues in your manuscript perhaps that could be de-emphasized to help with readability of the manuscript. Also some statement about what sample size might be needed since you already had a large sample available would be nice; in order to determine the feasibility of evaluating those hypotheses. Do you have any sense how many facilities are limiting access to joint replacement already based on these outcome scores which would add a potentially large confounder to your results. I think this article requires either an accompanying editorial or comment on a recommendation of the utility of using the outcomes scores as part of clinical treatment decision making. Thank you for your attention to detail and your thorough responses
--

REVIEWER	Bart Ferket Icahn School of Medicine at Mount Sinai, New York, NY, USA
REVIEW RETURNED	15-Dec-2017

GENERAL COMMENTS	I think the manuscript has much improved, I have no further comments.
---

VERSION 2 – AUTHOR RESPONSE

We thank you very much for your constructive comments and suggestions. The constructive critique helped us to improve the quality of the manuscript significantly. We believe that we were able to address all concerns and questions raised. Please find our responses to your points below.

~~~~~

Reviewer #3

Comment:

I think the authors have thoroughly revised the paper and accommodated reviewer comments. The additional tables are helpful in my view. I noted one typo - the mean and standard deviation for Annual costs of readmissions in years 1-6 for THR was missing a minus. I also thought that the title of table 2 might benefit from the addition of 'at primary operation' or something similar just to highlight that the OHS/OKS data is always for the primary procedure and not prior to revision. I'll leave that to the author's discretion.

Response:

Thank you very much for your comment. We apologise for the oversight. In the revised version of the manuscript, we have corrected the typo and amended the title of table 2 following your suggestions.

Reviewer #5

Comment:

This manuscript remains very dense reading although very thorough and thoughtful. I applaud the authors for not wanting to have multiple manuscripts but I think in this particular case two more separate manuscripts would help with the reading of the material; in particular one for clinicians and perhaps one for health economists. Part of the problem remains that you have a variety of places where you cannot draw conclusions due to the merging of data sets with different information and years.

Response:

Thank you very much for your comments as well as for your understanding. We appreciate that the manuscript presents a large number of analyses that may be of interest to different audiences. Nevertheless we would prefer to present the material in one manuscript to facilitate future use in modelling studies (as mentioned previously). Our other referees also appear to be content with the current structure. However, if the editors feel that it would be appropriate and acceptable to the journal and other referees, we would be prepared to divide the materials into two manuscripts.

Comment:

Your conclusion in the manuscript and the abstract should be the same.

Response:

We have amended the conclusion section of the abstract to better reflect the discussion in the manuscript.

Comment:

Based on all the variations of models etc a statement about your confidence in your findings might be helpful. If you were unable to prove the economics issues in your manuscript perhaps that could be de-emphasized to help with readability of the manuscript. Also some statement about what sample size might be needed since you already had a large sample available would be nice; in order to determine the feasibility of evaluating those hypotheses.

Response:

Thank you very much for your suggestion. We have added the following paragraph on p. 18/19 of the manuscript to discuss the extent to which the limitations affect the different findings presented in the manuscript. We also discuss how these shortcomings could be addressed in future research.

“These limitations affect our findings to varying degrees: There is very little uncertainty around the associations between OHS/OKS and quality-of-life as well as the cost of primary arthroplasty, since the underlying models were estimated on large samples. Similarly, our findings for the cost of hospital readmissions were based on large samples, however, the cost variables are more likely to be affected by the coding and classification errors mentioned above. While our findings on ambulatory costs and cost after revision arthroplasty are based on the best available datasets, there is still considerable uncertainty around these estimates due to relatively small sample sizes as well as coding and classification errors. These shortcomings could be addressed in future research, e.g., through the use of administrative datasets on primary care such as the Clinical Practice Research Datalink (CPRD) as well as data on revision arthroplasty from the National Joint Registry (NJR). Unfortunately, these datasets were not available for this research.”

Comment:

Do you have any sense how many facilities are limiting access to joint replacement already based on these outcome scores which would add a potentially large confounder to your results.

Response:

A 2014 report by the Royal College of Surgeons found that 16 out of 52 Clinical Commissioning Groups in the UK restricted access to total hip replacement based on an OHS threshold score. However, the report noted that this approach contradicted current guidelines, and that there was no consistency in the chosen threshold score. Moreover, the use of such threshold scores is likely to vary over time.

Consequently, we argue that this limitation is unlikely to affect our results substantially. We have amended the relevant part of the discussion section on p.18, which now reads as follows:

“A report commissioned by the Royal College of Surgeons in 2014 noted that 16 out of 52 Clinical Commissioning Groups (CCG) restricted access to THR based on an OHS threshold score.[6] However, current guidelines do not recommend a threshold for referral, and thus existing threshold scores are not based on evidence and they are not applied systematically. Therefore, these existing approaches should not have a major impact on our findings.”

Comment:

I think this article requires either an accompanying editorial or comment on a recommendation of the utility of using the outcomes scores as part of clinical treatment decision making.

Thank you for your attention to detail and your thorough responses

Response:

Thank you very much for your helpful comments and suggestions. The research reported in this manuscript was conducted as part of the larger ACHE study. The main clinical findings of this study, which build on the methodology and the results presented in this manuscript, will be submitted for publication soon. We agree that an accompanying editorial could help to orient and engage readers, but that of course is an editorial decision.